# THE CHALLENGE OF HIDDEN GIFTS IN MULTI-AGENT REINFORCEMENT LEARNING

## ABSTRACT

Sometimes we benefit from actions that others have taken even when we are unaware that they took those actions. For example, if your neighbor chooses not to take a parking spot in front of your house when you are not there, you can benefit, even without being aware that they took this action. These "hidden gifts" represent an interesting challenge for multi-agent reinforcement learning (MARL), since assigning credit when the beneficial actions of others are hidden is non-trivial. Here, we study the impact of hidden gifts with a very simple MARL task. In this task, agents in a grid-world environment have individual doors to unlock in order to obtain individual rewards. As well, if all the agents unlock their door the group receives a larger collective reward. However, there is only one key for all of the doors, such that the collective reward can only be obtained when the agents drop the key for others after they use it. Notably, there is nothing to indicate to an agent that the other agents have dropped the key, thus the act of dropping the key for others is a "hidden gift". We show that several different state-of-the-art MARL algorithms, including MARL specific architectures, fail to learn how to obtain the collective reward in this simple task. Interestingly, we find that decentralized actor-critic policy gradient agents can solve the task when we provide them with information about their own action history, but MARL agents still cannot solve the task with action history. Finally, we derive a correction term for these policy gradient agents, inspired by learning aware approaches, which reduces the variance in learning and helps them to converge to collective success more reliably. These results show that credit assignment in multi-agent settings can be particularly challenging in the presence of "hidden gifts", and demonstrate that self learning-awareness in decentralized agents can benefit these settings.

## 1 INTRODUCTION

In the world we often rely on other people to help us accomplish our goals. Sometimes, people help us even when we are not aware of it or haven't communicated a need for it. One simple example would be if someone decides not to take the last cookie in the pantry, leaving it for others. Another interesting example is the historical "Manitokan" practice of the plains Indigenous nations of North America. In an expansive environment with limited opportunities for communication, people would cache goods for others to use at effigies (Barkwell, 2015). Notably, in these cases there was no explicit agreement of a trade or articulation of a "tit-for-tat"(Axelrod, 1980). Rather, people simply engaged in altruistic acts that others could then benefit from, even without knowing who had taken the altruistic act. We refer to these undeclared altruistic acts as "hidden gifts".

Hidden gifts represent an interesting challenge for credit assignment in multi-agent reinforcement learning (MARL). If one leaves a hidden gift, assigning credit to the actions of another is essentially impossible, since the action was never made clear to the beneficiary. As such, standard Bellman-backups (Bellman, 1954) would likely be unable to identify the critical steps that led to success in the task. Moreover, unlike a scenario where cooperation and altruistic acts can emerge through explicit agreement or a strategic equilibrium (Nash Jr, 1950), as in general sum games (Axelrod, 1980), with hidden gifts the benefits of taking an altruistic action are harder to identify or reciprocate.

To explore the challenge of hidden gifts for MARL we built a grid-world task where hidden gifts are required for optimal behavior (Chevalier-Boisvert et al., 2023). We call it the Manitokan task, in

reference to the "take what you need, leave what you don't need" inspiration from Manitokan of plains Indigenous communities. In the Manitokan task, two-or-more agents are placed in an environment where each agent has a "door" that they must open in order to obtain an individual, immediate, small reward. As well, if all of the agents successfully open their door then a larger, collective reward is given to all of them. To open the doors, the agents must use a key, which the agents can both pick up and drop. However, there is only a single key in the environment. As such, if agents are to obtain the larger collective reward then they must drop the key for others to use after they have used it themselves. The agents receive an egocentric, top-down partial image of the environment as their observation in the task, and they can select actions of moving in the environment, picking up a key, dropping a key, or opening a door. Since the agents do not have access to other agent's decision making process, key drops represent a form of hidden gift – which make the credit assignment problem challenging. In particular: **1.** The task is fully cooperative so there is no disincentive for leaving the key, and **2.** dropping the key only leads to the collective reward if the other agents exploits the gift.

We tested several state-of-the-art MARL algorithms on the Manitokan task. Specifically we tested Value Decomposition Networks (VDN, QMIX and QTRAN) (Sunehag et al., 2017; Son et al., 2019; Rashid et al., 2020), Multi-Agent and Independent Proximal Policy Optimization (MAPPO and IPPO) (Schulman et al., 2017; Yu et al., 2022), counterfactual multi-agent policy gradients (COMA) (Foerster et al., 2018; She et al., 2022), Multi-Agent Variational Exploration Networks (MAVEN) (Mahajan et al., 2019), an information bottleneck based Stateful Active Facilitator (SAF) (Liu et al., 2023b) and standard actor-critic policy gradients (PG) with Actor-Critic (Williams, 1992; Sutton et al., 1999a; 1998; She et al., 2022). Notably, we found that none were capable of learning to drop the key and obtain the collective reward reliably. In fact, many of the MARL algorithms exhibited a total removal of key-dropping behavior, leading to less than random performance on the collective reward. These failures held even when we provided the agents with objective relevant information, providing inputs indicating which doors were open and whether the agents were holding the key.

Interestingly, when we also provided the agents with a history of their own actions as one-hot vectors, we observed that policy gradient agents without proximal policy optimization could now solve the collective task, whereas others still failed. However, these successful agents' showed high variability in cooperation. Based on this, we analyzed the value estimation problem for this task formally, and observed that the value function necessitates an approximation of a non-constant reward. That is, the collective reward is conditioned on the other agent's policy which is non-stationary between policy updates. Inspired by learning awareness (Willi et al., 2022; Foerster et al., 2017), we derived a new term in the policy gradient theorem which corresponds to the Hessian of the collective reward objective partitioned by the other agent's policy with respect to the collective reward. Using this correction term, we show that we can reduce the variance in the performance of the PG agents and achieve consistent learning to drop the key for others.

Altogether, our key contributions in this paper are:

- A structural credit assignment problem of hidden gifts induced in the Manitokan task.

- Evidence that several state-of-the art MARL credit assingment algorithms cannot solve the Manitokan task, even with recurrent policies, despite its small environment space.

- A demonstration that when action history is provided to recurrent PG agents, they can solve the task, while other algorithms still cannot.

- A theoretical analysis of the Manitokan credit assignment problem and a derived correction term inspired by learning-aware gradient updates (Foerster et al., 2017).

- A fully *decentralized* self learning-awareness term that does not require access to the other agent's policy, reduces variance and improves convergence towards leaving hidden gifts.

## 2 RELATED WORK

### 2.1 COORDINATION AND GIFTING IN MARL

Fully cooperative coordination games feature a single team objective requiring agents to act jointly, often reducible to a single-agent problem with a large action space. Previous tasks include navigation

(Mordatch & Abbeel, 2017; Lowe et al., 2017), cooking coordination (Carroll et al., 2019; Gessler et al., 2025), battles (Samvelyan et al., 2019; Ellis et al., 2023), and social-dilemmas (Leibo et al., 2017; Lerer & Peysakhovich, 2017; Christianos et al., 2020). These are often studied under the centralized training with decentralized execution, with methods such as COMA (Foerster et al., 2018) and QMIX (Rashid et al., 2020) leveraging global states during training to stabilize coordination. Additionally sharing collective rewards across agents is common and promotes cooperation but can also create "lazy-agent" credit assignment behavior (Liu et al., 2023a). Individualized rewards can mitigate this but risk pulling policies away from team objectives (Wang et al., 2022).

Within this cooperative context, "gifting" has been proposed as a mechanism for reward transfer, where one agent deliberately allocates part of its payoff to another to foster cooperation or reciprocity (Hughes et al., 2018; Peysakhovich & Lerer, 2018; Lupu & Precup, 2020). This can be seen as a bounded, targeted form of social influence. In single-agent RL this gifting can be interpreted as an intrinsic "self-gift," i.e., intrinsically generated rewards that support exploration or long-horizon credit assignment (Schmidhuber, 1991; Arjona-Medina et al., 2019; Sun et al., 2023). In multi-agent settings, intrinsic rewards have also been used to shape others' behavior through causal social influence (Jaques et al., 2019). However, this gifting is treated only as scalar reward signals, not as the transfer of tangible, task-critical resources.

## 2.2 MULTI-OBJECTIVE RL

Many decision-making problems involve objectives whose relative importance shifts over time, creating a non-stationary optimization landscape where fixed-weight multi-objective RL (MORL) methods falter (Van Moffaert & Nowé, 2014; Roijers et al., 2013). Dynamic-weights MORL addresses this by conditioning policies or value functions on the current weight vector $w(t)$, enabling a single policy to adapt across changing trade-offs without retraining. Approaches include weight-conditioned DQNs (Mossalam et al., 2016), policy gradients with weight inputs (Abels et al., 2019), and replay strategies for stability under shifting scalarizations (Yang et al., 2019).

In multi-agent settings, MORL has been used to balance individual and collective goals (Hayes et al., 2022), but prior work assumes known or designed $w(t)$, rather than treating another agent's policy itself as a dynamic weight. Seldom in the world do we have ever complete control of our incentives.

## 3 THE MANITOKAN TASK FOR STUDYING HIDDEN GIFTS

The Manitokan task is a cooperative MARL task in a grid world (see Fig.1). The task has been designed to be more complex than matrix games, such as Iterative Prisoner's Dilemma (Axelrod, 1980; Chammah, 1965), but capable for mathematical analysis of strategic behaviour and different from past cooperative environments (See 2). At the beginning of an episode each agent is assigned a locked door (Fig.1A) that they can only open if they hold a key. Agents can pick up the key if they move to the grid location where it is located (Fig.1B). Once an agent has opened their door it disappears and that agent receives a small individual reward immediately (Fig.1C). However, there is only one key for all agents to share and the agents can drop the key at any time if they hold it (Fig.1D). Once the key has been dropped the other agents can pick it up (Fig.1E) and use it to open their door as well (Fig.1F). If all doors are opened a larger collective reward is given to all agents, and at that point, the task terminates. The conditions for the rewards Eq. (1) are not mutually exclusive.

We now define the notation that we will use for describing the Manitokan task and analyzing formally. The environment is a decentralized partially observable Markov decision process (Dec-POMDP) with the caveat that the collective reward requires individual rewards (Goldman & Zilberstein, 2004; Bernstein et al., 2002). Dec-POMDPs are also a type of partially observable stochastic games (Hansen et al., 2004).

Let $M = (\mathcal{N}, T, \mathcal{T}, \mathcal{O}, \mathcal{A}, \Pi, \mathcal{R}, \gamma)$, where: $\mathcal{N} := \{1, 2, \ldots, N\}$ is the set of $N$ agents, $T \in \mathbb{N}$ is the maximum timesteps in an episode, $\mathcal{O} := \times_{i \in \mathcal{N}} O^i$ is the joint observation space for the $N$ agents and $o_t^i \in O^i \to \mathbb{N}^{3 \times 3}$ is a partial observation for an agent $i$ at timestep $t$. This is the only input agents take so the state $\mathcal{S} = \mathcal{O}$, $\mathcal{A} := \times_{i \in \mathcal{N}} A^i$ is the joint action space and $a_t^i \in A^i$ is the action of agent $i$ at time $t$, $\Pi := \times_{i \in \mathcal{N}} \pi^i$ is the joint space of individual agent policies, $\mathcal{R} \to \mathbb{R}$ is the reward function composed of both individual rewards, $r_t^i$, which agents receive for opening their own door (i.e. an individual objective), and the collective reward, $r^c$, which is given to all agents when all doors are

opened (i.e. a collective objective) (See equation 1 below.), $\mathcal{T} : \mathcal{O} \times \mathcal{A} \rightarrow \Delta(\mathcal{O})$ is the transition function specifying the probability $\mathcal{T}(o^{i'}, \mathcal{R}^i(o^i, a^i)|o^i, a^i)$ that agent $i$ transitions to $o^{i'}$ from $o^i$ by taking action $a^i$ for a reward $\mathcal{R}^i$, and $\gamma \in [0, 1)$ is the discount factor.

The observations, $o_t^i$, that each agent receives are egocentric images of the 9 grid locations surrounding the current position of the agent (see the lighter portions in Fig. 1). The key, the doors, and the other agents are all visible if they are in the field of view, but not otherwise (hence the task is partially observable). The actions the agents can select, $a_t^i$, consist of '*move forward*', '*turn left*', '*turn right*', '*pick up the key*', '*drop the key*', and '*open the door*'. Episodes last for $T = 150$ timesteps at maximum, and are terminated early if all doors are opened.

The monotonic reward function $\mathcal{R}^i$ is defined as:

$$\mathcal{R}^i(o_t^i, a_t^i) := \begin{cases} r_t^i = r^i & \text{door opened} \\ r^c = \sum_j^N r^j & \text{all doors opened} \end{cases} \tag{1}$$

But in correspondence with multi-objective problems, $\mathcal{R}^i$ is scalarized as $\hat{\mathcal{R}}^i = r_i + \omega(t)r^c$ where the preference weighting $\omega(t)$ is the other agent's policy so $\hat{\mathcal{R}}^i = r_i + {}^e\pi^j(a_t^j|o_t^j)r^c$ for agent $i$ and episode $e$ (Mossalam et al., 2016). The Manitokan task is unique from other credit assignment work in MARL due to the number of keys being strictly less than the number of agents (see. Section 2.1). This scarcity requires the coordination of gifting the key between agents as a necessary critical step for success and maximizing the cumulative return. But, notably, unlike most other MARL settings the act of dropping the key is not actually observable by other agents when learning a policy. When an agent picks up the key they do not know if they were the first agent to do so or if other agents had held the key and dropped it for them. Thus, key drop acts are "hidden gifts" between agents and the task represents a deceptively simple, but actually complex structural credit assignment problem across learning dynamics (Tumer et al., 2002; Agogino & Tumer, 2004; Gupta et al., 2021).

Importantly, the collective reward is delayed relative to any key drop actions. Moreover, key drop actions only lead to reward if the other agents have learned to accomplish their individual tasks. It then follows that the delay between a key drop action and the collective reward being received will be proportional in expectation to the number of agents, rendering a more difficult credit assignment problem for higher values of $N$. In the presented data, we focus on the canonical two-player setting from game theory, where ($N = 2$), for analytical tractability and interpretability of a Dec-POMDP.

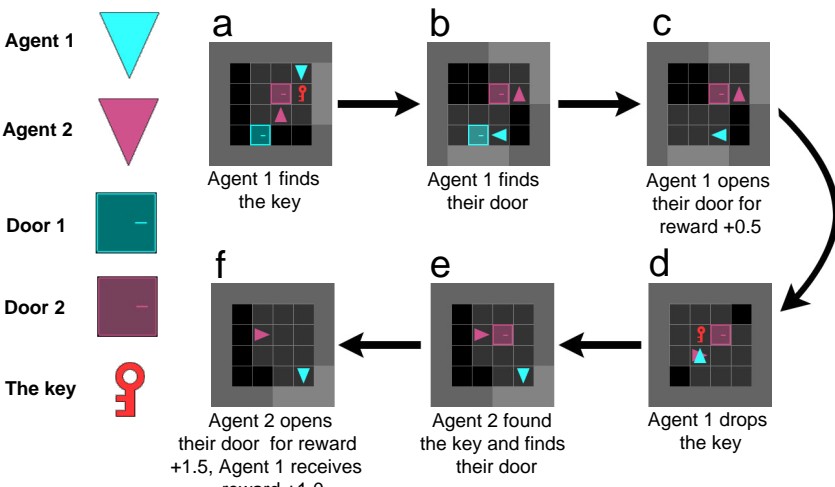

Figure 1: The deceivingly simple steps to success in the Manitokan task. a) Agent 1 finds the key; b) Agent 1 then finds their door; c) Agent 1 opens their door; d) Agent 1 drops the key as a "hidden gift"; e) Agent 3 finds their door; f) Agent 2 opens their door.

## 4 RESULTS

We begin by testing the ability of various state-of-the-art model-free RL algorithms to solve this task, both multi-agent, and decentralized. For the multi-agent algorithms, we selected ones that are prominently used as baselines for credit assignment in fully cooperative MARL tasks. These included the counterfactual model COMA, the centralized critic multi-agent PPO (MAPPO), and global value mixer algorithms VDN, QMIX and QTRAN (Foerster et al., 2018; Yu et al., 2022; Sunehag et al., 2017; Rashid et al., 2020; Son et al., 2019). We used actor-critic policy gradient methods, and gradient decoupled IPPO without a value function. (Williams, 1992; Sutton et al., 1999a; Schulman et al., 2017). In order to alleviate problems with exploration and changing policies we also tested MAVEN (which provides more robust exploration) and SAF (which is a meta-learning approach with a communication protocol network for learning with multiple policies) (Mahajan et al., 2019; Liu et al., 2023b). All algorithms were built with recurrent components in their policy (specifically, Gated Recurrent Units, GRUs (Cho et al., 2014)) in order to provide agents with some information about task history. (See methods in Appendix A for more details on design and training.) In our initial tests we provided only the egocentric (i.e agent's "self" is included) observations as input for the agents. Hyperparameters were optimized by tuning from the sets provided in the original papers with a search to avoid overfitting on the immediate reward. As well, we trained 10 simulations with different seeds that initialized 32 parallel environments also with different random seeds. These parallel environments make the reward signals in each batch less sparse. For each simulation we ran 10,000 episodes for each 32 parallel environments, except in Figure 5 where we did 26,000 episodes. Training was done with 2 CPUs for each run and SAF required an additional A100 GPU per run. An emulator was also used to improve environment step speed (Suarez, 2024).

### 4.1 ALL ALGORITHMS FAIL IN THE BASIC MANITOKAN TASK

To our surprise, everything we tested converged to a level of success in obtaining the collective reward that was *below* the level achieved by a fully random policy (Fig. 2a) even though reward was being maximized and the single agent key-to-door task is solvable (see E.8). In fact, with the sole exception of MAPPO, all of the MARL algorithms we tested (COMA, VDN, QMIX, QTRAN) exhibited full collapse in hidden gift behavior: these algorithms all converged to policies that involved *less* than random key dropping frequency. Randomizing the policy can slightly improve success rate but reduced cumulative reward ( 4). Notably, the agents that didn't show full collapse in collective success (MAPPO, IPPO and SAF) were still successfully opening their individual doors, since their cumulative reward was higher than that of a random policy (Fig. 2b). But, the MARL agents that showed total collapse of collective behavior also showed collapse in the individual rewards. We believe that this was due to the impact of asymmetric state information and shared value updates. With shared value updates the reward signal could be swamped by noise from the unrewarded agents in the absence of key drops, and became confused by a lack of reward obtained when agents' dropped the key before opening their doors (See more below in section 5). The key drop rate is optimal at 1, eg. one drop after using the key, all agents had a near zero drop rate or did not seem to learn (E.2).

### 4.2 OBSERVABILITY OF DOOR AND KEY STATUS DOES NOT RESCUE PERFORMANCE IN THE MANITOKAN TASK

To receive the collective reward, agents needed to learn to pick up the key, use it, then drop it. If they did these actions out of sequence (e.g. dropping the key before using it), then they can not succeed. As such, one potential cause for collapse in performance could have been the fact that agents did not have an explicit signal for their door being opened or that they are holding the key (i.e. the task was partially observable with respect to these variables). To make the task easier, we provided the agents with more decentralized information, one which indicated whether their door was open, the other which indicated whether they held the key. The agents now always have a cue when their individual task is completed.

Surprisingly, the agents we tested all failed to achieve collective success rates above random. In fact, the same behavior occurred, with the MARL agents (MAPPO, QMIX, COMA) showing total collapse, and the decentralized PG agents showing some collective success, but still below random (Fig. 3a). As before, We found that only MAPPO and decentralized PG showed any learning in the task, with QMIX and COMA showing collapse in the individual success rate as well (Fig. 3b).

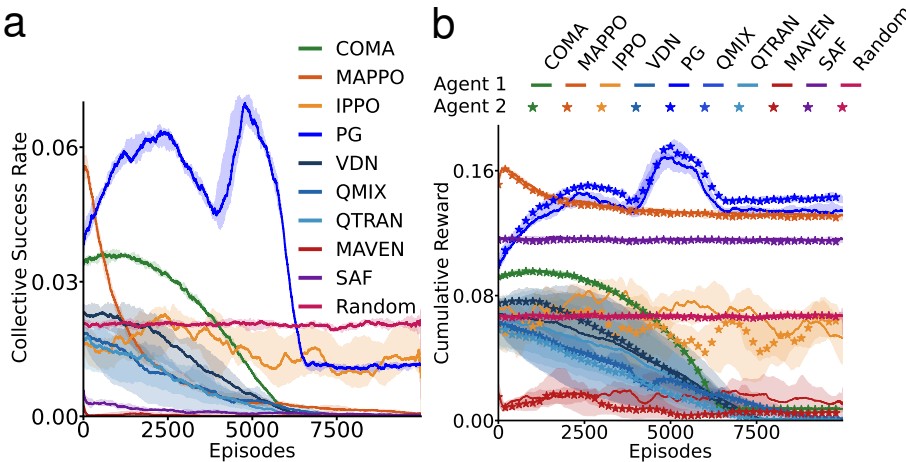

Figure 2: a) Success rate for the collective reward, i.e. percentage of trials where both agents opened their doors. b) Cumulative reward of both agents across 10000 episodes with 32 parallel environments limited to 150 timesteps each.

Thus, the lack of information about the status of the door and key was not the cause of failure of the Manitokan task.

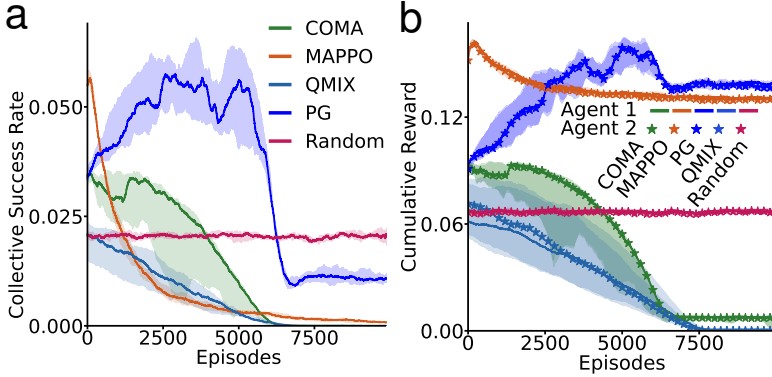

Figure 3: a) Success rate when each agent receives information about whether they have opened their door or not and if they have the key or not. b) Cumulative reward of both agents with information about whether they have opened their door or not and if they have the key or not.

### 4.3 ADDING ACTION HISTORY HELPS DECENTRALIZED AGENTS BUT NOT MARL AGENTS

Next, we reasoned that a cause of failure was that agents could not see themselves drop the key. To alleviate the credit assignment, we provided the agents with the last action they took as a one-hot vector. Coupled with the recurrence, this would permit the agents to know that they had dropped the key in the past if/when the collective reward was obtained.

When we added the past action to the observation, we found that the PG agents now showed signs of obtaining the collective reward, much better than random (Fig. 4). This also led to better cumulative reward (Fig. 4). However, interestingly, the other agents showed no ability to learn this task, exhibiting the same collapse in collective success rate and same low levels of cumulative reward as before (Fig. 4a & 4b). These results indicated that there is something about the credit assignment problem in the Manitokan task that can be addressed by the standard policy gradient objective, but not fancier trust region mechanisms. Further modifying the reward function can help or inhibit these agents but

removing one of the rewards harms success (E. 6). Changing or randomizing agent turn order also reduces success rate (E. 3). Overall, the PG agents still exhibited very high variance in their collective success rate (Fig. 4), suggesting more to the credit assignment problem. We then formally analyzed the value function of the task to better understand the credit assignment problem therein.

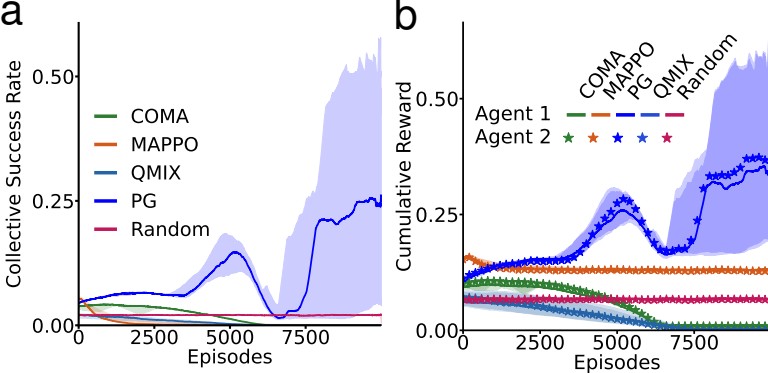

Figure 4: a) Success rate when each agent receives their last action in the observation. b) Cumulative reward of both agents with last action information.

## 5 FORMAL ANALYSIS AND CORRECTION TERM

For ease of Dec-POMDP analysis we again focus on the situation where $N = 2$, i.e. there are only two agents, and borrow the language of *sub-policies* from options learning (Sutton et al., 1999b). We begin by considering the objective function for agent $i$ with parameters $\Theta^i$, for an entire episode of the Manitokan task, where we ignore the discount factors (which do not affect the analysis) and take expectations over trajectories $\tau$ sampled from the policy $\pi$ of an agent given a randomly initialized observation:

$$J(\Theta^i) = \mathbb{E}_{\tau \sim \pi^i}[\sum_{t=0}^{T} \hat{\mathcal{R}}^i(o_t^i, a_t^i)] = \mathbb{E}_{\tau \sim \pi^i}[\sum_{t=0}^{T} r_t^i + r_t^c] = \mathbb{E}_{\tau \sim \pi^i}[\sum_{t=0}^{T} r_t^i] + \mathbb{E}_{\tau \sim \pi^i}[\sum_{t=0}^{T} r_t^c] \quad (2)$$

If we consider the sub-objective related solely to the collective reward $J_c(\Theta^i) = J(\Theta^i) - \mathbb{E}_{\tau \sim \pi^i}[\sum_{t=0}^{T} r_t^i] = \mathbb{E}_{\tau \sim \pi^i}[\sum_{t=0}^{T} r_t^c]$, we can then also consider the sub-policy of the agent related to the collective reward $(\pi_c^i)$, and the sub-policy unrelated to the collective reward $\pi_d^i$. If we condition the collective reward objective on the door for agent $i$ being open, then $J_c(\Theta^i)$ is independent of $\pi_d^i$. Therefore, when we consider the gradient for agent $i$ of the collective objective, conditioned on their door being open, we get:

$$\nabla_{\Theta^i} J_c(\Theta^i) = \mathbb{E}_{\tau \sim \pi^i} \left[ \nabla_{\Theta^i} \log \pi_c^i(a^i|o^i) \, Q_c(o^i, a^i) \right]$$
$$= \mathbb{E}_{\tau \sim \pi^i} \left[ \nabla_{\Theta^i} \log \pi_c^i(a^i|o^i) \right] \mathbb{E}_{\tau \sim \pi^i} \left[ Q_c(o^i, a^i) \right] \quad (3)$$

where $Q_c(o^i, a^i)$ is the value solely related to the collective reward. The gradient of this collective objective is inversely related to the entropy of the other agent's policy.

**Theorem 1.** *Let* $J_c(\Theta^i) = \mathbb{E}_{\tau \sim \pi^i}[\sum_{t=0}^{T} r_t^c]$ *be the collective objective function for agent* $i$, *and assume that agent* $i$ *is the first to open their door. Then the gradient of this objective function is given by:*

$$\nabla_{\Theta^i} J_c(\Theta^i) = \mathbb{E}_{\tau \sim \pi^j}[\nabla_{\Theta^i} \nabla_{\Theta^j} J_c(\Theta^j) \Psi(\pi_c^j, a^j, o^j)] \quad (4)$$

*where the element-wise reciprocal* $\Psi(\pi_c^j, a^j, o^j) = \mathbb{E}_{\tau \sim \pi^j}\left[\frac{1}{\nabla_{\Theta^j} \log \pi_c^j(a^j|o^j)}\right]$ *and* $i \neq j$.

See P.1 for the full proof. As a sketch, we rely on two key assumptions. The first key assumption is that agent $i$ is the first to open their door. As a result, agent $j$'s entire policy is related directly to the collective reward, and hence the sub-policy $\pi_d^j$ does not exist. The second key assumption is that the other agent's collective reward policy is differentiable. With those assumptions we can then use the objective of agent $j$ as a surrogate for the collective reward in the look-ahead step of the policy gradient derivation (Sutton et al., 1998), similar to mutual learning aware update rules (Willi et al., 2022; Foerster et al., 2017). The correction term does not conflict with individual objectives (see P.3) and is computed with a finite difference method. The complete gradient objective from P.1 is:

$$\nabla_{\Theta^i} J(\Theta^i) = \mathbb{E}_{\tau^i \sim \pi^i, \tau^j \sim \pi^j} [\nabla_{\Theta^i} \log \pi^i(a^i|o^i) Q(o^i, a^i) + \nabla_{\Theta^i} \nabla_{\Theta^j} J_c(\Theta^j) \Psi(\pi_c^j, a^j, o^j)] \quad (5)$$

### 5.1 USE OF A CORRECTION TERM IN THE VALUE FUNCTION

The correction Eq. (5) should reduce the variance in the agents' abilities to obtain the collective reward by stabilizing their value estimate with respect to each other's policies updating. Since the reward is shared, agents only need to correct with their own parameters in expectation (see proof in P. 2). This leads to a *decentralized* correction term of $\nabla_{\Theta^i} \nabla_{\Theta^i} J_c(\Theta^i) \Psi(\pi_c^i, a^i, o^i)$, which we term "Self Correction". Hence, we evaluated these policy gradient correction terms Fig.5.

With action history inputs, we trained PG agents with and without the correction and self-correction terms over seven days to ensure convergence. Additionally, we examined PG agents with a maximum entropy term, which should also reduce the variance in the learned policies (Ahmed et al., 2019; Haarnoja et al., 2018; Eysenbach & Levine, 2022). We found that all of the agents converged to a fairly high success rate over time (Fig. 5a), high cumulative reward (Fig. 5b) and reduced the distance between themselves (Fig.11b). Notably, the collective success variance was markedly different. The variance of the standard PG agents was quite high with steep spikes, and the variance of the max-entropy agents were not any different throughout the majority of the episodes, with the exception of the very early episodes (Fig. 5c). In contrast, the variance of the agents with the correction term was a bit lower but more stable. Interestingly, the agents with the self-correction term showed the lowest variance. We believe that this may be due to added noise from considering multiple policies in the update. Altogether, these results show that the correction term reduces variance in performance in the hidden gift problem, but is more prominent when decentralized with self-correction. This is interesting because it shows that it may be possible to resolve the complexities of hidden gift credit assignment using self learning-awareness rather than collective learning-awareness.

**Improved variance with the derived correction term**

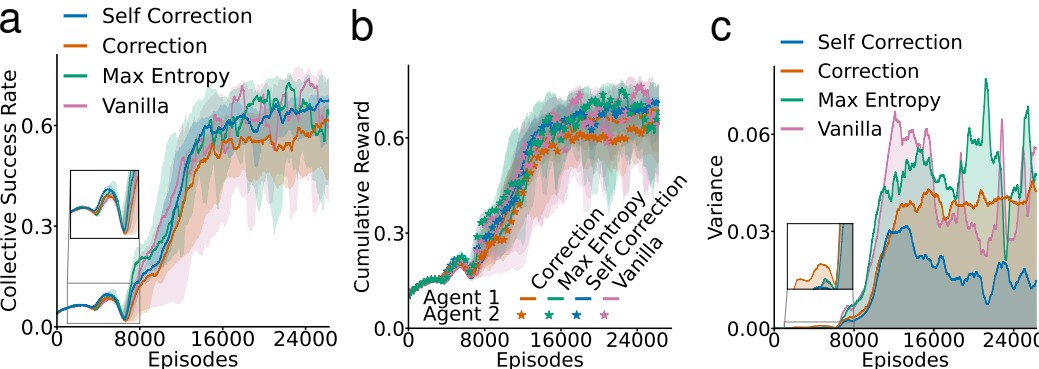

Figure 5: a) Success rate of PG agents comparing the vanilla PG model against PG with a maximum entropy term, PG with the correction term, and PG with the self-correction term. b) Cumulative reward of PG agents c) Variance in collective success rate across episodes.

## 6 DISCUSSION

In this work we developed a MARL task to explore the complexities of learning in the presence of "hidden gifts", i.e. cooperative acts that are not revealed to the recipient. The Manitokan task we developed, inspired by the concept in Indigenous plains communities across North America, requires agents to open doors using a single shared key in the environment. Agents must drop the key for other agents after they have used it if they are to obtain a larger collective reward. But, these key drop acts are not apparent to the other agents, making it difficult to assign credit between policy updates.

We observed that in the basic version of the Manitokan task none of the algorithms tested were able to solve it. This included both policy gradient agents (PG, PPO), meta-learning agents (SAF), enhanced exploration agents (MAVEN), counterfactual agents (COMA), and agents with collective value functions (VDN, QMIX, QTRAN, and MAPPO). When we added additional information to the observations the more sophisticated algorithms tested were still not able to solve this task. However, with previous action information, the actor-critic PG agents could solve the task, though with high variance. Formal analysis of the value function for the Manitokan task showed that it contains a second-order term related to the collective reward that can reduce instability in learning. We used this to derive a correction for the PG agents that successfully reduced the variance in their performance. Altogether, our results demonstrate that hidden gifts introduce challenging credit assignment problems that many state-of-the-art MARL architectures were not designed to overcome.

### 6.1 LIMITATIONS

We used a grid world task to induce the hidden gift credit assignment problem while enabling a tractable formal analysis. But, the real world is salient with sensory information and biological agents have large action spaces at their disposal. Less sparse signals and states, like the real world, may make the credit assignment problem easier with more information to leverage or infer.

Additionally, biological agents have a capacity for explicit, structured inter-agent communication which may aid in planning objectives or roles (Wu et al., 2024). This communication is different than the latent communication protocol of SAF (Liu et al., 2023b) in that agents could communicate to commit to gifting before hand which could become implicit and unspoken over time (Vélez et al., 2022). This may have been how similar practices developed in the plains of North America.

Lastly, the limited memory provided by the GRU architecture may inhibit credit assignment. It is possible that by integrating a more explicit form of memory with action history (e.g. a long context-window transformer), agents could more easily assign credit to their gifting behavior (Ni et al., 2023; Chen et al., 2021; Cross et al., 2025). A retrieval augmented temporal memory mechanism (Hung et al., 2019) might even help model-free agents avoid learning policies deviating away from or discounting the collective reward objective which may be non-markovian (Pitis, 2023). This *temporal* retrieval mechanism is different than SAF's *spatial* mechanism (Liu et al., 2023b).

### 6.2 RETHINKING RECIPROCITY

A broader implication from our work is that the emergence of reciprocity in a multi-agent setting can be complicated when acts of reciprocity themselves are partially or fully unobservable and therefore temporally indirect (Nowak & Sigmund, 2005; Santos et al., 2021). One potential interesting way of dealing with these situations would be to develop agents that are good at either predicting the actions of other agents or influencing other agents with implicit information (Jaques et al., 2019; Xie et al., 2021), which would ease the inference that other agents would exploit altruistic gifts. The reciprocity in MARL settings with any form of "hidden gift" may generally be aided by the ability of RL agents to successfully predict the actions of others when information is asymmetric. Given that the correction term that we derived from our formal analysis was motivated by the gradient steering effect in various learning aware approaches (Willi et al., 2022; Foerster et al., 2017; Meulemans et al., 2025; Aghajohari et al., 2024), it seems reasonable to speculate that abstracting properties from learning awareness, which may not always be optimal (see E. 9), have an untapped potential exterior to the domains in which they were designed. Such as inhibiting cooperation or collusion (see E.14).

**Reproducibility Statement**   The source code for all experiments and Manitokan task can be be found in the supplementary file along with a ReadMe file for setup. Hyperparameters used for the plots can be found in the methods Section. 1 of the appendix and modified in the config files of the source code. Hardware used and the time for experiments are mentioned in Section. 4 and described in the Methods section. 2 of the appendix. A sketch of the proof is found in the Section. 5 and the full derivations are found in the Proofs section of the appendix: P. 1, P. 2 and P. 3.

**Ethics Statement**   This work uses only synthetic simulation and did not use human participants nor any datasets; no personal data or sensitive attributes were collected. We recognize that increases in artificial intelligence performance or agentic capabilities may also increase negative societal risks (e.g., covert collusion or manipulation in multi-agent settings). However, the Manitokan task and the proposed correction term are explicitly designed to evaluate and improve RL agents' altruistic actions. We therefore do not foresee a specific safety concern from this work, especially relative to the alternative of RL agents that are not capable of taking altruistic actions. To reduce dual-use risk, we restrict claims to simulated environments and discourage harmful applications. No demographic or group attributes are used, and we aim for inclusive citation practices and accessibility. We acknowledge the cultural inspiration referenced by "Manitokan" and intend respectful use. A born and raised Indigenous community member, in the culture of which "Manitokan" existed, was deeply involve from beginning through the end of the work presented here.

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

# A  APPENDIX

# M METHODS

Methods  This section contains the hyperparameters for the results, hardware details for training and minor details on the task setup.

## M.1 HYPERPARAMETERS

Table 1: Model architecture and hyperparameters used for MAPPO.

| Component | Specification |
|---|---|
| Policy Network Architecture (Joint) | 1-layer CNN (outchannels = 32, kernel = 3, ReLU), 1-layer MLP (input = 32, output=64, ReLU), 1 layer MLP (input = 32, output=64, ReLU), 1 layer MLP (input = 64, output=64, ReLU), 1 layer GRU (input = 64, output = 64, with LayerNorm), 1 layer Categorical (input=64, output=6) |
| Value Network Architecture (Joint) | 1-layer CNN (outchannels = 32, kernel = 3, ReLU), 1-layer MLP (input = 32, output=64, ReLU), 1 layer MLP (input = 32, output=64, ReLU), 1 layer MLP (input = 64, output=64, ReLU), 1 layer GRU (input = 64, output = 64, with LayerNorm), 1 layer MLP(input = 64, output = 1, ReLU) |
| Optimizer | Adam, learning rate: $1 \times 10^{-5}$ |
| Discount Factor $\gamma$ | 0.99 |
| GAE Parameter $\lambda$ | 0.95 |
| PPO Clip Ratio $\epsilon$ | 0.2 |
| Entropy Coefficient | 0.0001 |
| Data chunk length | 10 |
| Parallel Environments | 32 |
| Batch Size | Parallel Environments × Data chunk length × number of agents |
| Mini-batch Size | 1 |
| Epochs per Update | 15 |
| Gradient Clipping | 10 |
| Value Function Coef. | 1 |
| Gain | 0.01 |
| Loss | Huber Loss with delta 10.00 |

Table 2: Model architecture and hyperparameters used for IPPO.

| Component | Specification |
|---|---|
| Policy Network Architecture (Disjoint) | 1-layer CNN (outchannels = 32, kernel = 3, ReLU), 1-layer MLP (input = 32, output=64, ReLU), 1 layer MLP (input = 32, output=64, ReLU), 1 layer MLP (input = 64, output=64, ReLU), 1 layer GRU (input = 64, output = 64, with LayerNorm), 1 layer Categorical (input=64, output=6) |
| Value Network Architecture (Disjoint) | 1-layer CNN (outchannels = 32, kernel = 3, ReLU), 1-layer MLP (input = 32, output=64, ReLU), 1 layer MLP (input = 32, output=64, ReLU), 1 layer MLP (input = 64, output=64, ReLU), 1 layer GRU (input = 64, output = 64, with LayerNorm), 1 layer MLP(input = 64, output = 1, ReLU) |
| Optimizer | Adam |
| Learning rate | $1 \times 10^{-5}$ |
| Discount Factor $\gamma$ | 0.99 |
| GAE Parameter $\lambda$ | Not used |
| PPO Clip Ratio $\epsilon$ | 0.2 |
| Entropy Coefficient | 0.0001 |
| Data chunk length | 10 |
| Parallel Environments | 32 |
| Batch Size | Parallel Environments × Data chunk length × number of agents |
| Mini-batch Size | 1 |
| Epochs per Update | 15 |
| Gradient Clipping | 10 |
| Value Function Coef. | 1 |
| Gain | 0.01 |
| Loss | Huber Loss with delta 10.00 |

Table 3: Model architecture and hyperparameters used for PG.

| Component | Specification |
| --- | --- |
| Policy Network Architecture (Joint) | 1-layer MLP (input = 27, output=64, ReLU), 1 layer GRU (input = 64, output = 64), 1 layer MLP (input=64, output=6) |
| Critic Network Architecture (Disjoint) | 1-layer MLP (input = 27, output=64, ReLU), 1-layer MLP (input = 64, output=64, ReLU), 1-layer MLP (input=64, output=1) |
| Target Critic Network Architecture (Disjoint) | 1-layer MLP (input = 27, output=64, ReLU), 1-layer MLP (input = 64, output=64, ReLU), 1-layer MLP (input=64, output=1) |
| Actor optimizer | RMSprop, alpha $0.99$, epsilon $1 \times 10^{-5}$ |
| Critic optimizer | RMSprop, alpha $0.99$, epsilon $1 \times 10^{-5}$ |
| Discount factor $\gamma$ | 0.99 |
| Target network update interval | 1 episode |
| Learning rate | $5 \times 10^{-5}$ |
| TD Lambda | 1.0 |
| Replay buffer size | 32 |
| Parallel environment | 32 |
| Parallel episodes per buffer episode | 1 |
| Training batch size | 32 |

Table 4: Model architecture and hyperparameters used for COMA.

| Component | Specification |
| --- | --- |
| Policy Network Architecture (Joint) | 1-layer MLP (input = 27, output=64, ReLU), 1 layer GRU (input = 64, output = 64), 1 layer MLP (input=64, output=6) |
| Critic Network Architecture (Joint) | 1-layer MLP (input = 27, output=64, ReLU), 1-layer MLP (input = 64, output=64, ReLU), 1-layer MLP (input=64, output=6) |
| Target Critic Network Architecture (Joint) | 1-layer MLP (input = 27, output=64, ReLU), 1-layer MLP (input = 64, output=64, ReLU), 1-layer MLP (input=64, output=6) |
| Actor optimizer | RMSprop, alpha $0.99$, epsilon $1 \times 10^{-5}$ |
| Critic optimizer | RMSprop, alpha $0.99$, epsilon $1 \times 10^{-5}$ |
| Discount factor $\gamma$ | 0.99 |
| Target network update interval | 1 episode |
| Learning rate | $5 \times 10^{-5}$ |
| TD Lambda | 1.0 |
| Replay buffer size | 320 |
| Parallel environment | 32 |
| Parallel episodes per buffer episode | 1 |
| Training batch size | 32 |

Table 5: Model architecture and hyperparameters used for SAF.

| Component | Specification |
|---|---|
| Policy Network Architecture (Disjoint) | 2-layer MLP (input = 64, output=128, Tanh), |
| Value Network Architecture (Joint) | 2-layer MLP (input = 80, output=128, Tanh), |
| Shared Convolutional Encoder (Joint) | 1-Layer CNN (outchannels = 64, kernel = 2) |
| Knowledge Source Architecture (Joint) | |
|    Query Projector | 1-layer MLP (input = 128, output=64, Tanh) |
|    State Projector | 1-layer MLP (input = 128, output=64, Tanh) |
|    Perceiver Encoder | (latents = 4, latent input = 64, cross attention channels = 64, cross attention heads = 1, self attention heads = 1, self attention blocks = 2 with 2 layers each) |
|    Cross Attention | (heads = 1, query input = 64, key-value input = 64, query-key input = 64, value channels = 64, dropout = 0.0) |
|    Combined State Projector | 1-layer MLP (input = 128, output=64, Tanh) |
|    Latent Encoder | 1-layer MLP (input = 128, output=64, Tanh), 1-layer MLP (input = 64, output=64, Tanh ),1-layer MLP (input = 64, output=16, Tanh ) |
|    Latent Encoder Prior | 1-layer MLP (input = 64, output=64, Tanh), 1-layer MLP (input = 64, output=64, Tanh ),1-layer MLP (input = 64, output=16, Tanh ) |
|    Policy Projector | 1-layer MLP (input = 128, output=164, Tanh) |
| Optimizer | Adam, epsilon $1 \times 10^{-5}$ |
| learning rate | $3 \times 10^{-4}$ |
| Discount Factor $\gamma$ | 0.99 |
| GAE Parameter $\lambda$ | GAE not used |
| PPO Clip Ratio $\epsilon$ | 0.2 |
| Entropy Coefficient | 0.01 |
| Data chunk length | 10 |
| Parallel Environments | 32 |
| Batch Size | Parallel Environments $\times$ Data chunk length $\times$ number of agents |
| Mini-batch Size | 5 |
| Epochs per Update | 15 |
| Gradient Clipping | 9 |
| Value Function Coef. | 1 |
| Gain | 0.01 |
| Loss | Huber Loss with delta 10.00 |
| Number of policies | 4 |
| Number of slot keys | 4 |

Table 6: Model architecture and hyperparameters used for VDN.

| Component | Specification |
|---|---|
| Policy Network Architecture (Joint) | 1-layer MLP (input = 27, output=64, ReLU), 1 layer GRU (input = 128, output = 64), 1 layer MLP (input=128, output=6) |
| Target Policy Network Architecture (Joint) | 1-layer MLP (input = 27, output=64, ReLU), 1 layer GRU (input = 128, output = 64), 1 layer MLP (input=128, output=6) |
| Mixer Network Architecture | Tensor sum of states |
| Policy optimizer | Adam, alpha 0.99, epsilon $1 \times 10^{-5}$ |
| Target policy optimizer | Adam, alpha 0.99, epsilon $1 \times 10^{-5}$ |
| Start epsilon greedy | 1.0 |
| Minimum epsilon greedy | 0.05 |
| Discount factor $\gamma$ | 0.99 |
| Target network update interval | 1 episode |
| Start learning rate | $1 \times 10^{-2}$ |
| Minimum learning rate | $1 \times 10^{-6}$ |
| TD Lambda | 1.0 |
| Replay buffer size | 1000 |
| Parallel environment | 32 |
| Parallel episodes per buffer episode | 32 |
| Training batch size | 32 |
| Warm up buffer episodes | 32 |

Table 7: Model architecture and hyperparameters used for QMIX.

| Component | Specification |
| --- | --- |
| Actor Network Architecture (Joint) | 1-layer MLP (input = 27, output=64, ReLU), 1 layer GRU (input = 64, output = 64), 1 layer MLP (input=64, output=6) |
| Target Actor Network Architecture (Joint) | 1-layer MLP (input = 27, output=64, ReLU), 1 layer GRU (input = 64, output = 64), 1 layer MLP (input=64, output=6) |
| Mixing Network Architecture (Joint) | |
| Hypernet Weights 1 | 1-layer MLP (input =54, output=64, ReLU), 1-layer MLP (input = 64, output=52) |
| Hypernet Biases 1 | 1-layer MLP (input =54, output=64) |
| Hypernet Weights 2 | 1-layer MLP (input =54, output=32, ReLU), 1-layer MLP (input = 64, output=32) |
| Hypernet Bias 2 | 1-layer MLP (input =54, output=64, ReLU), 1-layer MLP (input = 64, output=1) |
| Target Mixing Network Architecture (Joint) | |
| Hypernet Weights 1 | 1-layer MLP (input =54, output=64, ReLU), 1-layer MLP (input = 64, output=52) |
| Hypernet Biases 1 | 1-layer MLP (input =54, output=64) |
| Hypernet Weights 2 | 1-layer MLP (input =54, output=32, ReLU), 1-layer MLP (input = 64, output=32) |
| Hypernet Bias 2 | 1-layer MLP (input =54, output=64, ReLU), 1-layer MLP (input = 64, output=1) |
| Policy optimizer | Adam, alpha 0.99, epsilon $1 \times 10^{-5}$ |
| Target policy optimizer | Adam, alpha 0.99, epsilon $1 \times 10^{-5}$ |
| Start epsilon greedy | 1.0 |
| Minimum epsilon greedy | 0.05 |
| Discount factor $\gamma$ | 0.99 |
| Target network update interval | 1 episode |
| Start learning rate | $1 \times 10^{-2}$ |
| Minimum learning rate | $1 \times 10^{-6}$ |
| TD Lambda | 1.0 |
| Replay buffer size | 1000 |
| Parallel environment | 32 |
| Parallel episodes per buffer episode | 32 |
| Training batch size | 32 |
| Warm up buffer episodes | 32 |

Table 8: Model architecture and hyperparameters used for QTRAN.

| Component | Specification |
|---|---|
| Policy Network Architecture (Joint) | 1-layer MLP (input = 27, output=64, ReLU), 1 layer GRU (input = 64, output = 64), 1 layer MLP (input=64, output=6) |
| Target Policy Network Architecture (Joint) | 1-layer MLP (input = 27, output=64, ReLU), 1 layer GRU (input = 64, output = 64), 1 layer MLP (input=64, output=6) |
| Mixing Network Architecture (Joint) | |
|    Query Network | 1-layer MLP (input =188, output=32, ReLU), 1-layer MLP (input = 32, output=32, ReLU), 1-layer MLP (input = 32, output=1) |
|    Value Network | 1-layer MLP (input =54, output=32, ReLU), 1-layer MLP (input = 32, output=32, ReLU), 1-layer MLP (input = 32, output=1) |
|    Action Encoding | 1-layer MLP (input =134, output=134, ReLU), 1-layer MLP (input = 134, output=134) |
| Target Mixing Network Architecture (Joint) | |
|    Query Network | 1-layer MLP (input =188, output=32, ReLU), 1-layer MLP (input = 32, output=32, ReLU), 1-layer MLP (input = 32, output=1) |
|    Value Network | 1-layer MLP (input =54, output=32, ReLU), 1-layer MLP (input = 32, output=32, ReLU), 1-layer MLP (input = 32, output=1) |
|    Action Encoding | 1-layer MLP (input =134, output=134, ReLU), 1-layer MLP (input = 134, output=134) |
| Policy optimizer | Adam, alpha 0.99, epsilon $1 \times 10^{-5}$ |
| Target policy optimizer | Adam, alpha 0.99, epsilon $1 \times 10^{-5}$ |
| Start epsilon greedy | 1.0 |
| Minimum epsilon greedy | 0.05 |
| Discount factor $\gamma$ | 0.99 |
| Target network update interval | 1 episode |
| Start learning rate | $1 \times 10^{-2}$ |
| Minimum learning rate | $1 \times 10^{-6}$ |
| TD Lambda | 1.0 |
| Replay buffer size | 1000 |
| Parallel environment | 32 |
| Parallel episodes per buffer episode | 32 |
| Training batch size | 32 |
| Warm up buffer episodes | 32 |

Table 9: Model architecture and hyperparameters used for MAVEN.

| Component | Specification |
|---|---|
| Policy Network Architecture (Joint) | 1-layer MLP (input = 27, output=64, ReLU), 1 layer GRU (input = 64, output = 64), 1 layer MLP (input=64, output=6) |
| Target Policy Network Architecture (Joint) | 1-layer MLP (input = 27, output=64, ReLU), 1 layer GRU (input = 64, output = 64), 1 layer MLP (input=64, output=6) |
| Noise Mixing Network Architecture (Joint) | |
|    Hypernet Weights 1 | 1-layer MLP (input=116, output=64) |
|    Hypernet Bias 1 | 1-layer MLP (input=116, output=32) |
|    Hypernet Weights 2 | 1-layer MLP (input=116, output=32) |
|    Skip Connection | 1-layer MLP (input=116, output=2) |
|    Value network | 1-layer MLP (input=116, output=32, ReLU), 1-layer MLP(input=32,output=1) |
| Target Noise Mixing Network Architecture (Joint) | |
|    Hypernet Weights 1 | 1-layer MLP (input=116, output=64) |
|    Hypernet Bias 1 | 1-layer MLP (input=116, output=32) |
|    Hypernet Weights 2 | 1-layer MLP (input=116, output=32) |
|    Skip Connection | 1-layer MLP (input=116, output=2) |
|    Value network | 1-layer MLP (input=116, output=32, ReLU), 1-layer MLP(input=32,output=1) |
| RNN Aggregator | 1-layer GRU (input=116, output=2) |
| Discriminator | 1-layer MLP (input=116, output=32, ReLU), 1-layer MLP (input=32, output=2), |
| Actor optimizer | RMSprop, alpha 0.99, epsilon $1 \times 10^{-5}$ |
| Target actor optimizer | Adam, alpha 0.99, epsilon $1 \times 10^{-5}$ |
| Use skip connection in mixer | False |
| Use RNN aggregation | False |
| Discount factor $\gamma$ | 0.99 |
| Target network update interval | 1 episode |
| Learning rate | $5 \times 10^{-5}$ |
| TD Lambda | 1.0 |
| Replay buffer size | 1000 |
| Parallel environment | 32 |
| Parallel episodes per buffer episode | 1 |
| Training batch size | 32 |

## M.2   COMPUTE

For each simulation 2 CPUs were allocated and the 32 parallel environments were multithreaded. All algorithms expect for SAF were able to run without GPUs while SAF used a single A100 for each simulation. All algorithms, except for VDN, QMIX and QTRAN can finish at 10000 episodes for all 10 simulations within 4 days while the aforementioned algorithms take 7 days. It is possible to use a GPU for these value mixer mechanisms for faster data collection but this was not done to collect the data. The correction term experiments take 7 days to collect 26000 episodes and do not benefit from GPUs since their networks are too small. The Hessian term can be approximated with finite difference technique or with Pearlmutter's trick.

## M.3   MANITOKAN TASK SETUP

The Manitokan Task is a grid world for tractable analysis. The key, agents and doors are randomly initialized at the beginning of each episode and the actions *drop* and *toggle* were additionally pruned when an agent is not holding a key for reasonable environment logic but are not necessary to be removed for the task to work. The doors look the same to both agents. Everything else was described in 3.

# E ADDITIONAL EXPERIMENTS

The experiments provided below offer insights into the challenge of the Manitokan Task, and further empirical validation of the correction and self correction terms.

## E.1   COMA'S LOSS BECOMES NEGATIVE

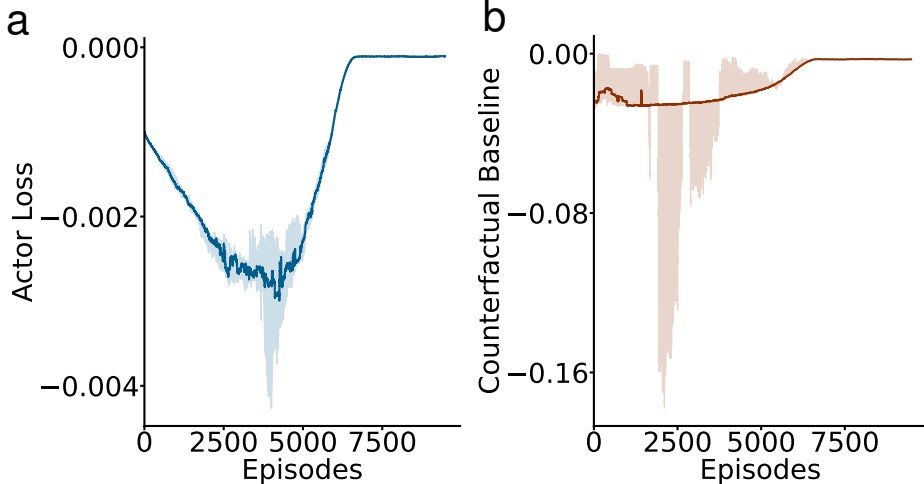

Figure 6: a) Policy loss of the COMA model b) Counterfactual baseline in the COMA policy update

COMA persistently collapsed even though it exhibited similar learning behaviour to PG (a closely related model). The policy loss and baseline curves show increasing instability with large variance spikes before converging to a value around 0.0. Perhaps this collapse is from the difficulty of leaving a hidden gift between individual and collective incentives. The original COMA paper (Foerster et al., 2018) even mentions a struggle for an agent overcoming an individual reward, although exterior to hidden gifts, may be cause for the instability.

## E.2 OPTIMAL KEY DROP RATE IS UNATTAINED BY ALL AGENTS

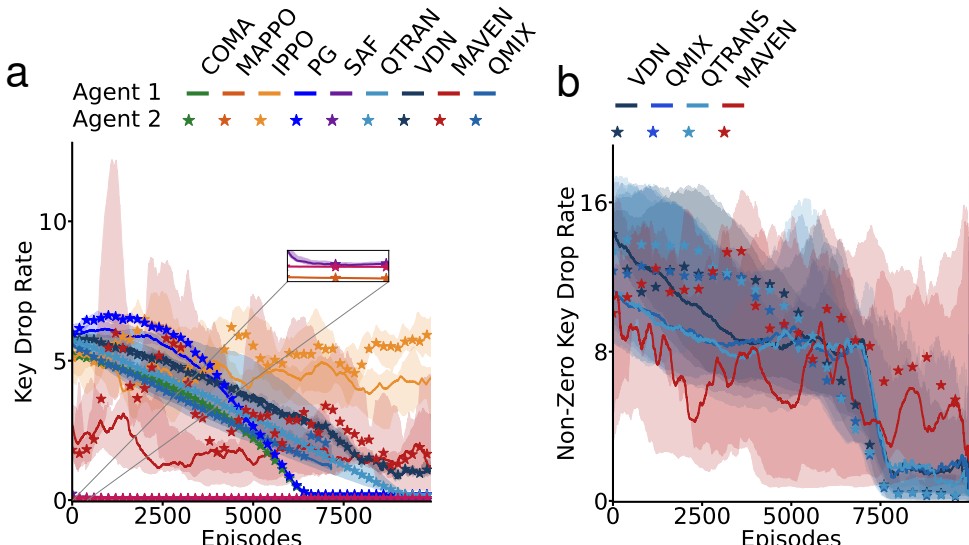

Figure 7: a) Key drop rate (i.e. cumulative key drops) averaged across parallel episodes and runs. b) Non-zero key drop rate (i.e. cumulative key drops) averaged across parallel episodes that had key drops and runs.

For most of the MARL agents (VDN, QMIX, QTRAN, MAVEN) the key drop rate always converged to exactly zero (Fig. 7), hence the total collapse in collective success in the task. In the case of MAPPO, and SAF, we observed that the agents learned to pick up the key and open their individual doors, but minimized the number of key drops to close to zero (Fig. 7a). As a result, the collective success rate was also close to zero. In contrast, IPPO did not exhibit a collapse in key drops but had an oscillatory effect where one would agent increase their keydrops while the other reduces theirs. This explains IPPO's slightly better success in obtaining the collective reward (Fig. 2a). Interestingly, COMA and decentralized PG showed very low, but non-zero rates of key drop (Fig. 7a), however only PG exhibited a non-zero collective success rate (Fig. 2a). This was because even though COMA agents learned to occasionally drop the key, the counter-factual baseline caused the loss to become excessively negative (see E.1).

One complication with measuring the key drop rate is that if the agents never even pick up the key then the key drop rate is necessarily zero. To better understand what was happening in here, we examined the "non-zero key drop rate", meaning the rate at which keys were dropped if they were picked up. The non-zero key drop rate showed that the value mixer MARL agents begin by dropping the key after picking it up some of the time, but eventually converge to a policy of simply holding or avoiding the key (Fig. 7b). The variance in drop rates is increased except at the end for VDN, QMIX and QTRAN. This further emphasizes the challenge of hidden gifts.

### E.3 CHANGING WHICH AGENT STEPS FIRST IN AN EPISODE HARMS PERFORMANCE

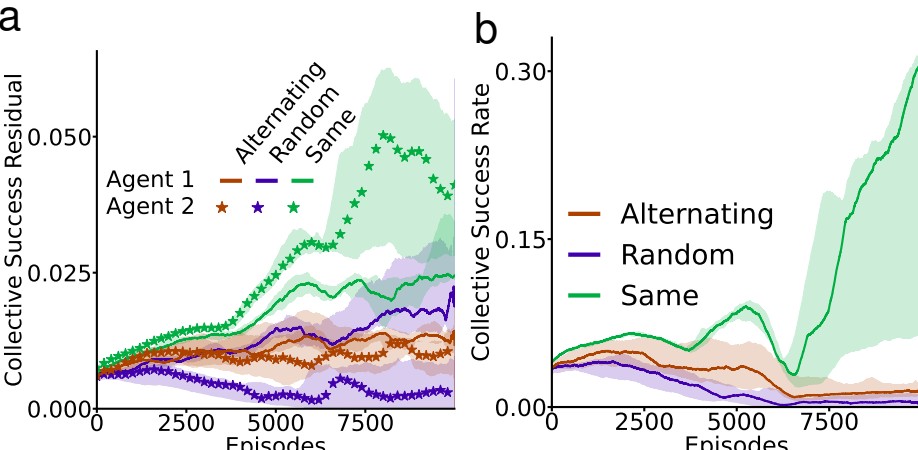

Figure 8: a) The contribution of an agent's reward accumulation to success weighted by their total reward comparing policy gradient agents with action history of the same agent stepping first (i.e. agent 1 then agent 2), alternating agents stepping first (i.e. agent 1 steps first on odd numbered episodes and agent 2 steps first in even numbers episodes), and a random agent is selecting to step first. b) Success rate between different step ordering each episode.

The collective success residual is calculated as $(r^c - r^i) \times r^i$ where $(r^c - r^i)$ describes how much an agent $i$ is contributing to the collective success while weighting it by $r^i$ shows if the agents are increasing that success rate. Interestingly, alternating which agent goes first between episodes creates oscillations in the collective success rate residual where one agent receiving more reward means the other agent receives less. Greatly reducing the success. Moreover, randomly selecting an agent to go first biases the first agent to increase their reward and almost removes all success. These effect may be caused by uncertainty associated with which agent can reach the key when the other agent is in sight. For example in the random case, if agent $i$'s current policy has learned for the past five updates that it will pick up the key, when both agents are equal distance from the key, there will be a action prediction error. This uncertainty increases the difficulty of the credit assignment problem.

## E.4 RANDOMIZING THE POLICY CAN SLIGHTLY INCREASE COLLECTIVE SUCCESS SLIGHTLY

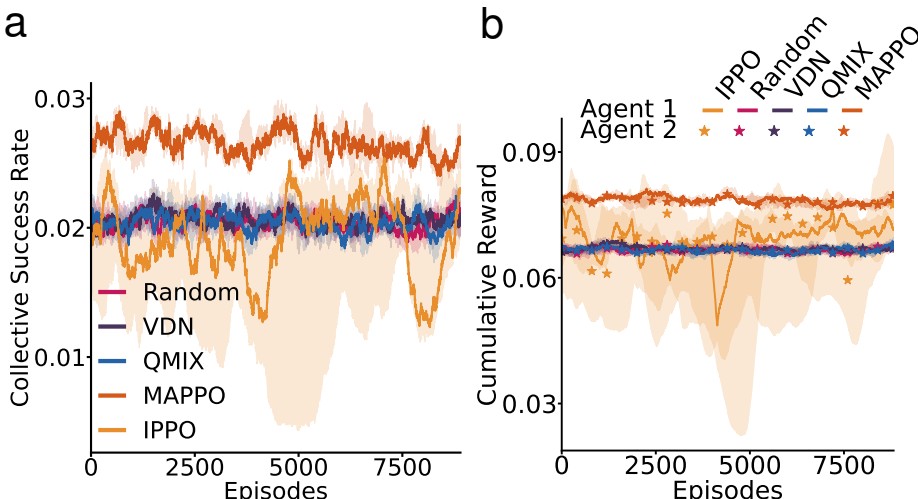

Figure 9: a) Comparing agents of MAPPO, IPPO, VDN and QMIX algorithms with a randomization applied to their policies b) The cumulative reward for randomized policy agents

PPO agents had their value function learning rates set to 0.001 while the policy learning rates where kept as 0.000001. This meant the policy would always prefer initial episodes and converge quickly to those while the value function weighting them more evenly to converge further in the training process. VDN and QMIX use epsilon greedy in their strategy and simply increasing the time of decay for this mechanism led these agents to be more random throughout the experiment.

This policy randomization process very slightly improved these agents the success rates' compared to those in the main results Fig 2a but decreased the cumulative reward for the PPO agents than those in Fig 2b. The random policy aligned VDN and QMIX to the random action baseline more or less, and avoided collapse.

## E.5 BEHAVIOURAL VARIATIONS APPEAR BETWEEN ALGORITHMS WITH INTER AGENT DISTANCE AND MINIMIZING THE STEPS TO THE FIRST REWARD

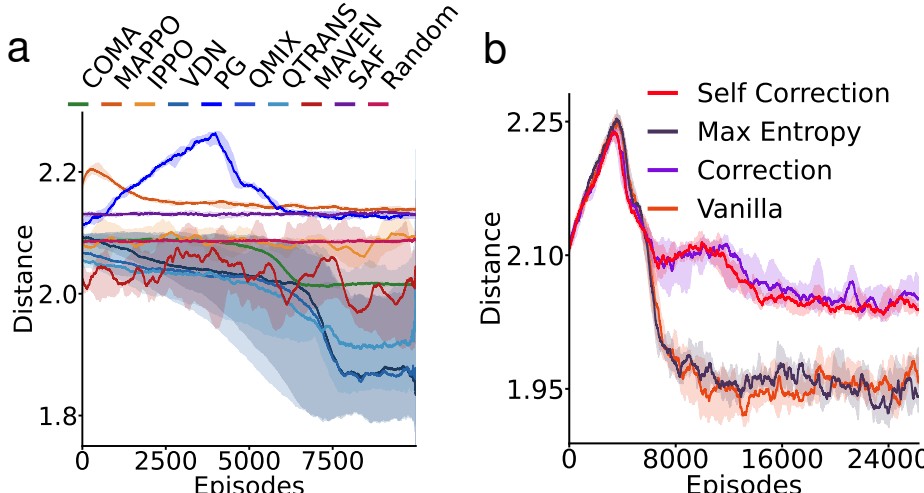

Figure 10: a) Euclidean distance between agents averaged over parallel environments and simulations across our tested models b) Euclidean distance comparing policy gradient agents with action history and variance reduction terms.

Although the 2-agent Manitokan Task is a four by four grid world, we measured the euclidean distance between agents to see if they become more coordinated or adversarial when learning hidden gifting. In Fig 10a, PG agents exhibited the highest exploration phase but eventually converged to a lower distance. MAPPO agents also has a similar but substantially smaller exploration effect in the very beginning while SAF did not have any exploration phases. IPPO and MAVEN agents similarly hovered below the random baseline but MAVEN agents were closer to each other. COMA agents begin around random but converge to be closer to each other as well. Value mixer agents VDN, QMIX and QTRAN all are on average closer to each other but QTRAN agent agents converge further apart.

In Fig 10b, vanilla and max entropy PG agents with action history become asymptotically closer to each other while the correction term agents converge further apart from them. The variance reduction in self correcting agents is also noticeable.

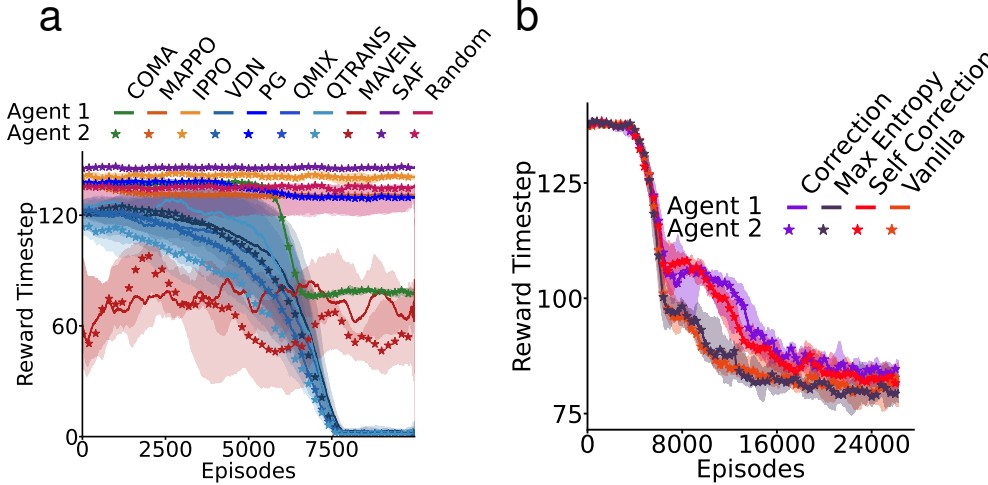

Figure 11: a) Timestep the first reward an agent received. b) Timestep the first reward a policy gradient agent with action history received.

The reducing the timestep of the first reward is a way to measure if agents are improving their policies if cumulative reward also increases. In (Fig 11a), PG, IPPO, MAPPO and SAF all converge quickly while PG and MAPPO learn policies of reducing the step slightly below random. COMA converges at a low timestep but this is most likely due to the collapse. MAVEN oscillates at a timestep better than random but never converges and doesn't seem to learn a good policy and VDN, QMIX, and QTRAN collapse consistently with other results in Section 3.

While in Fig 11b, all decentralized PG algorithms with action history reduce their initial reward timesteps but models with the correction term converge slower.

E.6    MODIFYING THE REWARD FUNCTION ENHANCES PERSPECTIVE ON THE CHALLENGE OF THE MANITOKAN TASK

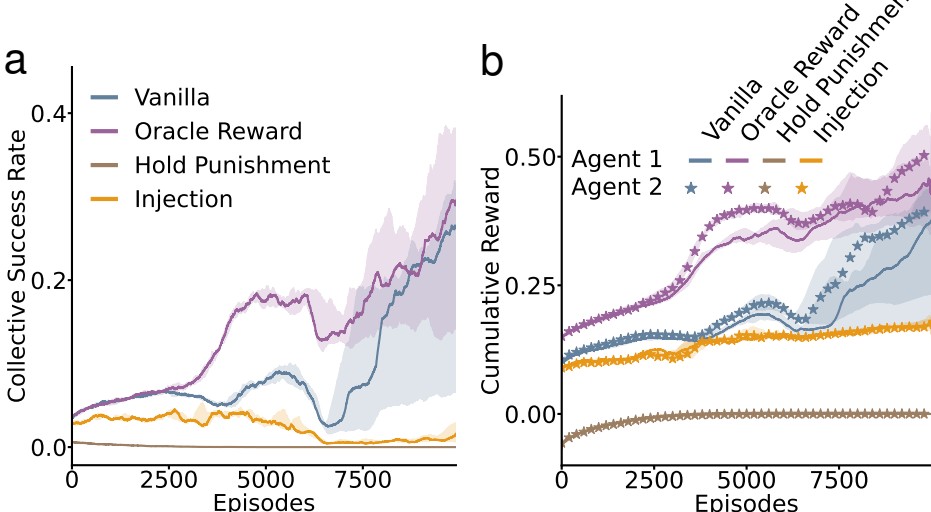

Figure 12: a) Success rate of policy gradient agents with action history comparing the normal reward function with an oracle reward term (i.e. an agent receives a reward of 1 once for dropping the key after opening their door), a punishment term (i.e.. a negative reward of 1 is applied each step an agent holds their key after opening their door) and a reward injection term (i.e. randomly distributing normally smaller rewards around the standard rewards decaying over episodes) b) Cumulative reward to compare the modified reward functions

The reward function $\mathcal{R}$ in equation 1 to study hidden gifting behavior is both sparse with a hard to predict collective reward conditioned on the other agent's policy. We tested additional reward conditions on PG agents with action history to see if sample efficiency improvement can be found. Particularly, the oracle reward: $r_t^i$ the first key dropped after agent $i$'s door is opened , is the critical step to take for hidden gifting and when implemented the collective success rate increased quicker than the normal reward function. The punishment reward: $-0.5$ for each step agent $i$ is holding the key after their door was opened, is also meant to induce gifting behavior but agents seemed to avoid the key altogether. Lastly, the injection reward where a set of rewards $r^d < r^i$ are normally distributed around rewards $r^i$ and $r^c$ which also served as the mean. $r^d$ was additionally reduced each episode for agents to prefer the standard rewards. Injection reduced the success rate severely but also reduced variance in accumulating the expected reward.

These minor modifications reemphasize the difficulty in hidden gifting, where our most performative agents still struggle even when rewarded for the optimal action.

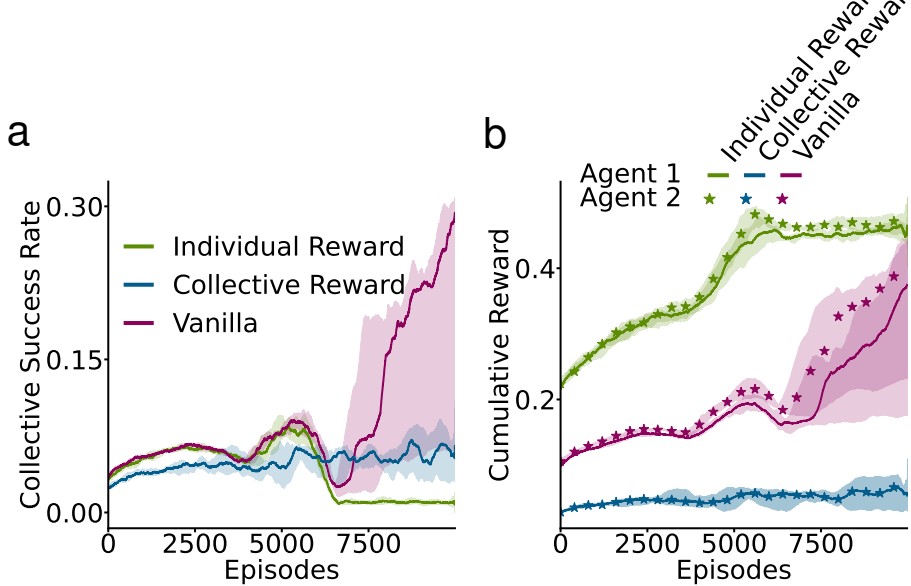

Figure 13: a) Success rate between policy gradient agents comparing a disassociation of the reward function (i.e.. just the individual reward and the collective rewards) b) Cumulative reward of the same dissociated reward function agents

For a further investigation of the reward function, we tested a dissociation of the individual reward $r^i$ and the collective reward $r^c$ with action history PG agents. Using only the individual reward, removed collective success altogether but agents converged at a higher percentage of the cumulative reward (i.e.. whoever gets to the key first). This is essentially an equilibrium with 50% probability of getting a reward. Isolating collective reward and removing the individual reward did not cause a failure in collective behavior but severely inhibited it. The success rate average did increasingly ossiclate. With both these reward dissociation, agents fail to learn hidden gifting.

### E.7 THE SELF CORRECTION TERM IS EMPIRICALLY SOUND IN CONTRAPOSITION

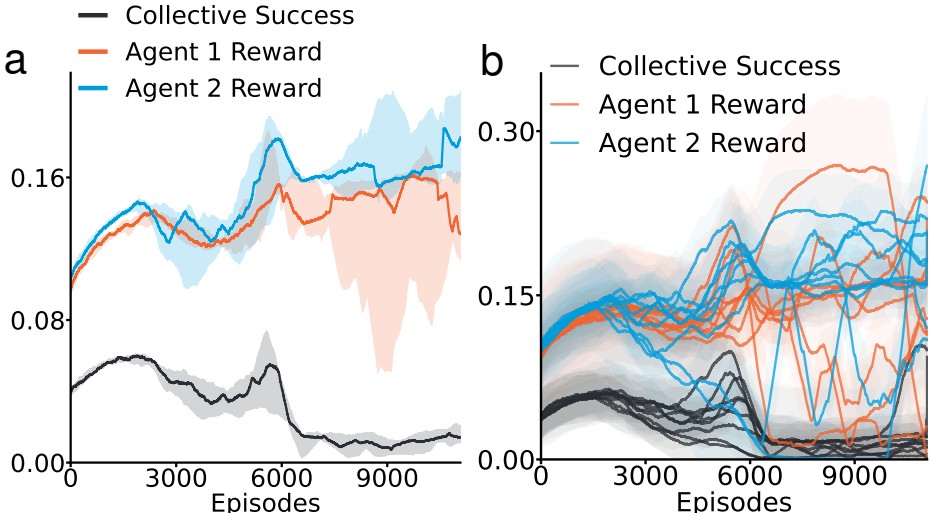

Figure 14: a) The percentage of cumulative reward and collective success for anti-collective policy gradient agents with action history (i.e. optimizing the negated self correction term) across 11000 episodes b) 9 individual simulations for anti-collective behaviour averaged each across a different set of 32 parallel environments

For all previous experiments, the correction term was maximized to induce agents towards dropping the key for the other agent (i.e. hidden gifting). Contrapositively however, this term for an agent $i$ could also be minimized through negation $-\mathbb{E}[\nabla_{\Theta^i}\nabla_{\Theta^j}J_c(\Theta^j)\Psi(\pi_c^j, a^j, o^j)]$ in the policy update and doing so led agents to actively "compete" for the key and avoid dropping it all together. In Fig 14a, the rewards for both agents increases with variance spikes while the collective success rate goes down. These results demonstrate a stronger implication of the self-correction in the collective behaviour of agents than just as a variance reducer.

Fig 14b displays the individual simulations with standard deviation of the 32 parallel environments. Specifically, the reward curves sharply drop and return after agents have learned to open their doors. This tradeoff in the individual reward accumulation is a detriment to the collective success rate but perhaps in other situations, the negative correction term can help avoid undesired rewarded behaviour.

Alternatively, if we set $\Psi$ to be $\frac{1}{\mathbb{E}[\nabla_{\Theta^j}\pi_c^j(a^j|o^j)]}$ instead of $\frac{1}{\mathbb{E}[\nabla_{\Theta^j}\log\pi_c^j(a^j|o^j)]}$, the self-correction term is now weighted by the actual collective policy rather than it's entropy. This is refereed to as $\hat{\Psi}$. This is a plug in adjustment and did not have a theoretical motivation or derivation. The policy independence from 1 should still hold but there is no proof for this adjustment. However in looking at Fig. 15, there is a different change in behaviour of the agents.

In Fig. 15a, the agents follow a very similar reward accumulation path but sharply drop around 3000 episodes where a slight switch in agent 1 achieving a higher reward. This happens again at a slower rate at around 7000 episodes where agent 2 accumulates more reward than the other agent and eventually surpassing agent 1 with some increase in variance until the end of the experiment. Agent 1's reward accumulation deteriorates after 8000 episodes implying that agent 2 is better at finding the key and always holding onto it. Then this happens for a final time in this experiment at 16000 episodes where the variance blows up. The negated self-correction here is inhibiting agents more sharply, perhaps due to the smaller range of values that $\pi_c(a|o)$ has than $\log\pi_c(a|o)$. The success rate is severely reduced and does not pickup in variance or on average through the remainder of the experiment.

In Fig. 15b, all simulations are plotted with their within simulation variance. The behaviour between both agents in performance and variance is similar until 6000 episodes where agent 2 over takes agent 1 in all but three simulations. This then reverses around 16000 episodes with the majority of agent 1

simulations overtaking agent 2. The success rate simulations are near identical to eachother, showing how more impactful $\hat{\Psi}$ is to inhibiting cooperation.

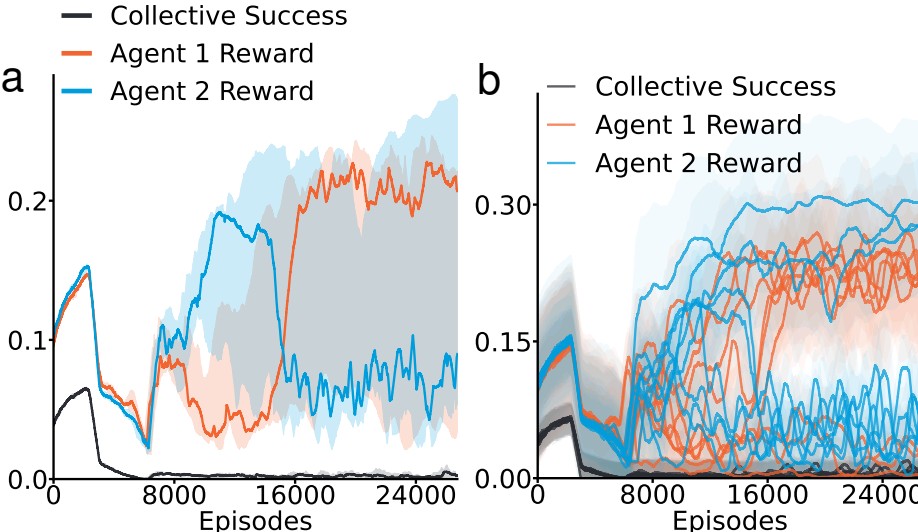

Figure 15: a) The percentage of cumulative reward and collective success for anti-collective policy gradient agents with action history (i.e. optimizing the negated self correction term) across 11000 episodes b) 9 individual simulations for anti-collective behaviour averaged each across a different set of 32 parallel environments

The original self-correction contraposition is theoretically motivated and showed a more smoother slower inhibition on average where agents continued to compete with similar performance until almost the end of the experiment. The adjusted self-correction inhibition has a more sharper effect which makes sense since the policy is a categorical distribution. The agents do not compete comparatively though. Agent 2 overtakes agent 1 earlier than in the first inhibition experiment. Overall, these experiments further implicate self-correction in learning the collective sub policy for leaving hidden gifts for the other agent.

## E.8 THE POLICY GRADIENT OBJECTIVE IS BETTER THAN THE Q-LEARNING IN SINGLE AGENT KEY-TO-DOOR

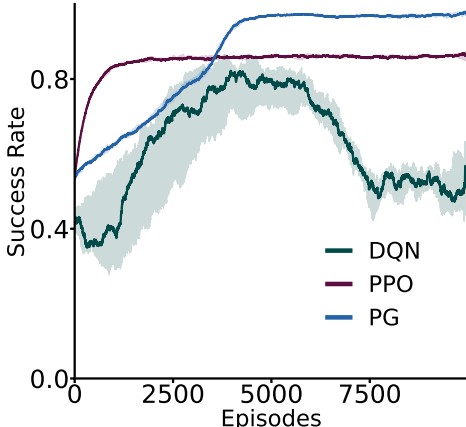

Figure 16: a) Comparison of single agent PPO, PG and DQN agents where one agent needs to open a one door after finding one key

As a baseline, PPO, PG and DQN agents are compared on the individual objective of the main task (eg. opening a door). This is a normal key-to-door task and success is defined by opening a door for a reward of 1. PPO and PG agents retain the same hyperparameter except the learning rate for both actor and critic in PPO was reduced after a grid search to tune against overfitting. The DQN agent required 1 simulation at a time rather than 32 in parallel but was not able to converge above 50% success after an extensive hyperparameter search. This demonstrates the performance of on-policy policy gradient objective over the off-policy q-learning objective in temporal credit assignment tasks.

### E.9 SELF-CORRECTION OUT PERFORMS LOLA ON THE MANITOKAN TASK

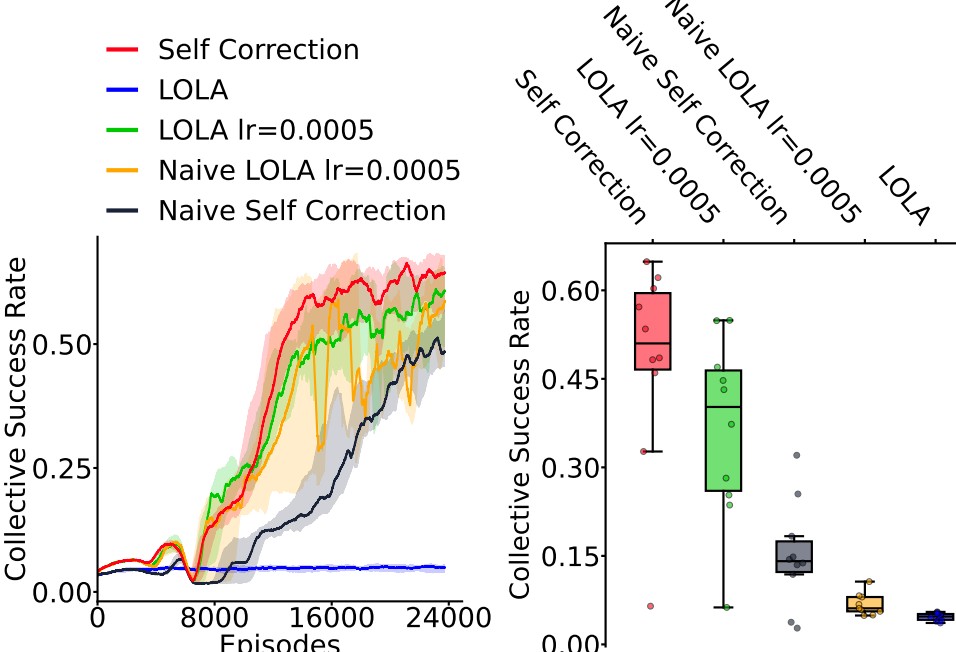

Figure 17: a) A comparison of LOLA with a learning rate of 1 and 0.0005, Naive LOLA with a learning rate of 0.0005, Self-correction, and Naive Self-correction. The naive learner framework only permits one agent to optimize the additional hessian objective rather than two. b)

Learning with Opponent Learning Awareness (LOLA) (Foerster et al., 2017) is the original learning aware gradient update. In the original work, only one policy gradient agent was a LOLA agent with the other being a naive policy gradient agent. To test how self-correction compares, 32 parallel simulations were ran for LOLA with a learning rate of 1 and 0.0005, Naive LOLA with a learning rate of 0.0005, Self-correction, and Naive Self-correction. LOLA with a learning rate of 1 did not learn cooperative behaviour but decreasing the learning rate to 0.0005 improved learning but with high ossicilations in median performance as well as variance. When both agents were LOLA agents, similar to (Willi et al., 2022), there was greater stability in collective success but less than self-correction. For thoroughness, a naive learner experiment for self-correction was ran. Interestingly, the variance reduction effect was maintained but performance was delayed and reduced compared to self-correction. However, this implies that one self-correction agent can help stabilize collective success.

E.10 EMPIRICAL ANALYSIS ON THE EFFECT ON THE INFORMATION OF LAST ACTIONS

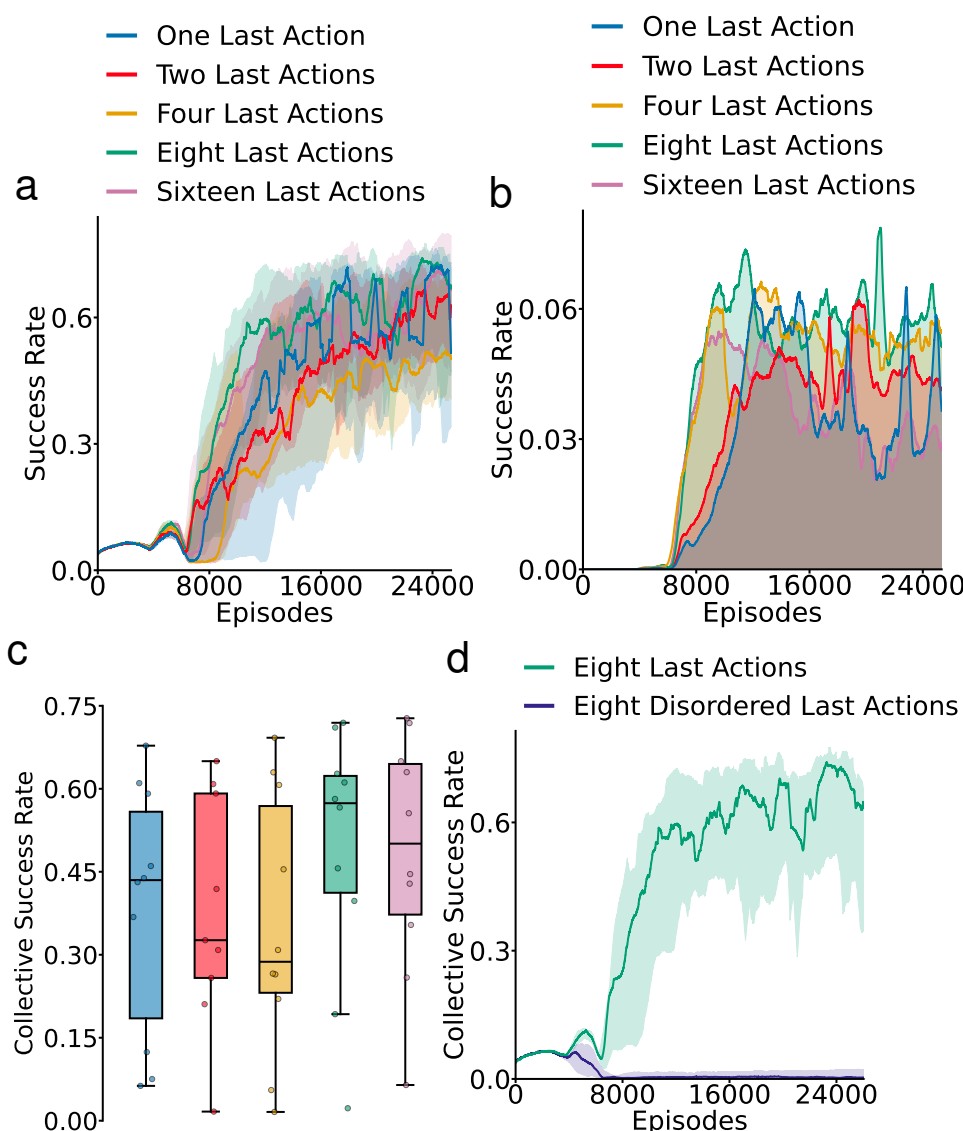

Figure 18: a) A comparison of collective success rate between PG agents with last action inputs with one last action, two last actions, four last actions, eight last actions and sixteen last actins b) A comparison of variance over time between PG agents with last action inputs c) A comparison of global collective success rate between PG agents with last action inputs d) A comparison between PG agents with eight last actions as input and PG agents with randomly permutated eight last actions as input.

Due to the performance of one last action, we were curious if more last actions could further increase the collective success rate. Two and Four last actions inhibit performance, while eight and sixteen last actions recover performance comparable to one last action. These results indicate that a single previous action provides a great deal of the necessary information to solve the task, though more history could also help. There did not seem to be a large effect on variance over time; however, sixteen last actions did exhibit a reduction. The importance of temporal structure rather than quantity of information is demonstrated in Fig. 18d where randomly permuting or disordering 8 last actions nearly brings collective success rate to zero.

### E.11   THE SELF CORRECTION VALUE IS MORE CORRELATED TO COLLECTIVE SUCCESS THAN POLICY ENTROPY IN MAXIMUM ENTROPY POLICY GRADIENTS

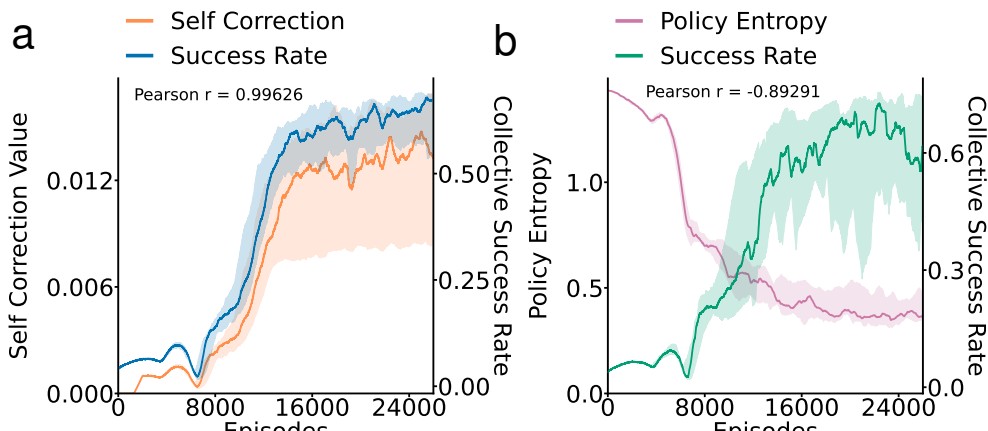

Figure 19: a) A correlation analysis between the collective success rate rate of PG agents with one last action input and a self correction term, and the value of the self-correction term. b) A correlation analysis between the collective success rate rate of PG agents with one last action input and a maximum entropy term, and the value the value of policy entropy.

In Fig. 19a, we tested the relationship between the value of the self correction term and our performance metric of collective success rate. The self correction term is highly correlated with success, since Pearson's r = 0.99626. The variance in the self-correction term is noticeably larger than the success rate. While in In Fig. 19b we did the same test between the policy entropy values and the collective success rate of the maximum entropy PG agents. The entropy is inversely correlated with the collective success rate but not as strongly than as the self correction values with a Pearson's r = -0.89291. The variance in policy entropy is markedly low.

### E.12   POLICY GRADIENT AGENTS WITH SELF CORRECTION'S COLLECTIVE SUCCESS RATE IS GLOBALLY LARGER THAN OTHER POLICY GRADIENT AGENTS

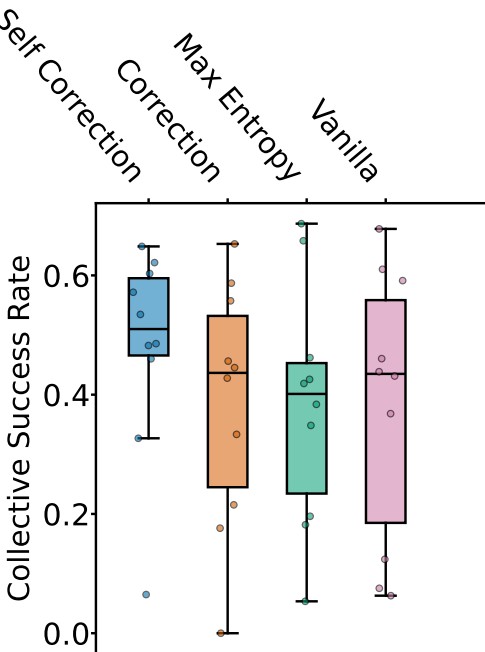

Figure 20: The comparison of global success rate across PG agents with self correction, correction and maximum entropy, as well as PG agents without any extra terms

Markedly, the median of the self correction term's global collective success rate is larger than other PG models. This speaks to how the stability provided by the correction term, not only reduces variance but also stabilizes collective success. The median of the normal correction term the second largest, slightly above the vanilla PG agents' collective success rate. The maximum entropy term had the lowest collective success rate out of the four PG models.

# P PROOFS

## P.1 CORRECTION TERM

We begin by deriving the standard policy gradient theorem (Sutton et al., 1998; 1999a) under the assumptions in Section 4 that an agent $i$ is first to open their door and that the collective reward $r^c$ is differentiable through another agent $j$s objective. The objective $J(\Theta^i)$ for agent $i$ is to maximize the expected cumulative sum of rewards within an episode $\mathbb{E}[\sum_t^T \mathcal{R}^i(o_t^i, a_t^i)]$ with the reward function $\mathcal{R}$ in equation 1 where a value function $V(\Theta^i, o^i) = \mathbb{E}[\mathcal{R}^i(o^i, a^i)]$.

$$\nabla_{\Theta^i} J(\Theta^i) = \nabla_{\Theta^i} \Big( \sum_{a^i \in A} \pi^i(a^i|o^i) Q(o^i, a^i) \Big) \tag{6}$$

is the differentiated objective with respect to agent $i$.

$$\sum_{a^i \in A} \Big( \nabla_{\Theta^i} \pi^i(a^i|o^i) Q(o^i, a^i) + \pi^i(a^i|o^i) \nabla_{\Theta^i} Q(o^i, a^i) \Big) \tag{7}$$

by product rule expansion.

$$\sum_{a^i \in A} \nabla_{\Theta^i} \pi^i(a^i|o^i) Q(o^i, a^i) + \pi^i(a^i|o^i) \nabla_{\Theta^i} \Big( \sum_{o^{i'}, R^i} \mathcal{T}(o^{i'}, R^i(o^i, a^i)|o^i, a^i)(R^i(o^i, a^i) + V(\Theta^i, o^{i'}) \Big) \tag{8}$$

Here, Eq. (8) is summed over all actions $\sum_{a^i \in A}$. In particular, the value function can be used to predict a look-ahead of the next reward with a next observation $o^{i'}$ and $\mathcal{T}$ is the transition probability.

Now we construct the other agent's value estimate as a surrogate for the future collective reward. The individual reward is a constant and disappears by passing the gradient but we can isolate the collective reward as sub-objective for a sub-policy with a linearity assumption.

$$\mathbb{E}_{\tau \sim \pi^j}\Big[\sum_{t=0}^T \hat{\mathcal{R}}^j(o_t^j, a_t^j)\Big] = \mathbb{E}_{\tau \sim \pi^j}\Big[\sum_{t=0}^T r_t^j + r_t^c\Big] = \mathbb{E}_{\tau \sim \pi^j}\Big[\sum_{t=0}^T r_t^j\Big] + \mathbb{E}_{\tau \sim \pi^j}\Big[\sum_{t=0}^T r_t^c\Big] \tag{9}$$

Eq. (1), only $r^j$ degenerates to 0 while $r^c$ is differentiable w.r.t to another agent $j$.

To isolate the sub-objective for the collective policy, start with the reward maximization objection.

$$J(\Theta^j) = \mathbb{E}_{\tau \sim \pi^j}\Big[\sum_t^T \hat{\mathcal{R}}^j(o_t^j, a_t^j)\Big] \tag{10}$$

$$J(\Theta^j) = \mathbb{E}_{\tau \sim \pi^j}\Big[\sum_{t=0}^T r_t^j\Big] + \mathbb{E}_{\tau \sim \pi^j}\Big[\sum_{t=0}^T r_t^c\Big] \tag{11}$$

by linearity in Eq. (2) of $\mathcal{R}^j$.

$$J(\Theta^j) - \mathbb{E}_{\tau \sim \pi^j}\Big[\sum_{t=0}^{T} r_t^j\Big] = \mathbb{E}_{\tau \sim \pi^j}\Big[\sum_{t=0}^{T} r_t^c\Big] = J_c(\Theta^j) \tag{12}$$

$$\nabla_{\Theta^j} J_c(\Theta^j) = \mathbb{E}_{\tau \sim \pi^j}[\nabla_{\Theta^j} \log \pi_c^j(a^j|o^j) Q_c(o^j, a^j)] \tag{13}$$

$$\nabla_{\Theta^j} J_c(\Theta^j) = \mathbb{E}_{\tau \sim \pi^j}[\nabla_{\Theta^j} \log \pi_c^j(a^j|o^j)] \mathbb{E}_{\tau \sim \pi^j}[Q_c(o^j, a^j)] \tag{14}$$

Since the individual policy on finding the key and opening the door is assumed to be learned from Eq. (3) . Now the collective policy of agent $j$ is probabilistically independent from that agent's collective Q-value $Q_c$ since the collective reward can only be acquired by agent $i$ who has it's own policy. There is no action agent $j$ can take to acquire the collective reward or improve it's $Q_c$ estimate after dropping the key.

Now to truly use $Q_c$ as a surrogate for the collective reward, we have to perform an element-wise division with $\mathbb{E}_{\tau \sim \pi^j}[\nabla_{\Theta^j} \log \pi_c^j(a^j|o^j)]$ but, the score function of a stochastic policy generally tends towards zero in expectation $\mathbb{E}_{\tau \sim \pi}[\nabla_\Theta \log \pi(a|o)] = 0$ because there is at least one optimal action $^*a^j$ to take at every observation $o^j$. So, minimizing the policy gradient objective increases the probability of the optimal actions in expectation.

However, there is no action $a^j$ the agent can take to acquire the collective reward $r^c$. So, every action $a^j$ under the softmax policy $\pi_c^j(a^j|o^j)$ is equally as likely as any other action. Therefore, actions are *uniformly* distributed in the collective policy $\pi_c^j$ which is by consequence the collective policy outputs a uniform distribution itself $\pi_c^j(; |o^j) = a^j \sim U(0, |\mathcal{A}|)$.

The gradient of the logarithm of a uniform distribution and it's expected are common statistical knowledge, and they are not rederived here. The maximum likelihood estimate of the collective policy is

$$\nabla_{\Theta^j} \log \pi_c^j(; |o^j) = -\frac{1}{|\mathcal{A}|} \neq 0 \tag{15}$$

and therefore, the gradient of the score function in expectation is also not zero

$$\mathbb{E}_{\tau \sim \pi^j}[\nabla_{\Theta^j} \log \pi_c^j(; |o^j)] = \mathbb{E}_{\tau \sim \pi^j}\Big[-\frac{1}{|\mathcal{A}|}\Big] \neq 0 \tag{16}$$

Let $\Psi(\pi_c^j, o^j, a^j) = \frac{1}{\mathbb{E}_{\tau \sim \pi^j}[\nabla_{\Theta^j} \log \pi_c^j(a^j|o^j)]}$ where $\Psi$ is the element wise reciprocal of the expected collective policy for agent $j$. So we can clarify the term

$$\frac{\nabla_{\Theta^j} J_c(\Theta^j)}{\mathbb{E}_{\tau \sim \pi^j}[\nabla_{\Theta^j} \log \pi_c^j(a^j|o^j)]} = \nabla_{\Theta^j} J_c(\Theta^j)\Psi(\pi_c^j, o^j, a^j) = \mathbb{E}_{\tau \sim \pi^j}[Q_c(o^j, a^j)] \tag{17}$$

$$\sum_{a^i \in A} \pi^i(a^i|o^i)\Big(\sum_{o^{i'}, \hat{\mathcal{R}}^i} \mathcal{T}(o_{+1}^{i'}, \hat{\mathcal{R}}^i(o^i, a^i)|o^i, a^i)(\nabla_{\Theta^i}\nabla_{\Theta^j} J_c(\Theta^j)\Psi(\pi_c^j, a^j, o^j) + \nabla_{\Theta^i} V(\Theta^i, o^{i'}))$$
$$\tag{18}$$

Now in Eq. (18) the correction term as a surrogate for the collective reward in the look ahead step from Eq. (8).

Let $\Phi(o^i) = \sum_{a^i \in A}(\nabla_{\Theta^i}\pi^i(a^i|o^i)Q(o^i,a^i)$ for readability and Let $\rho^i(o^i \to o^{i'}) = \pi^i(a^i|o^i)(\sum_{o^{i'},\hat{\mathcal{R}}^i}\mathcal{T}(o^{i'},\hat{\mathcal{R}}^i(o^i,a^i)|o^i,a^i)$ for further readability.

$$\Phi(o^i) + \sum_{o^i}\rho^i(o^i \to o^i_{+1})(\nabla_{\Theta^i}V(\Theta^i,o^i_{+1}) + \nabla_{\Theta^i}\nabla_{\Theta^j}J_c(\Theta^j)\Psi(\pi_{\Theta^j},a^j,o^j)) \tag{19}$$

The previous, Eq. (19), can then be recursively expanded out further $\Phi(o^i) + \sum_{o^i}\rho^i(o^i \to o^i_{+1})(\Phi(o^i_{+1}) + \nabla_{\Theta^i}\nabla_{\Theta^j}J_c(\Theta^j)\Psi(\pi^j_c,a^j,o^j) + \sum_{o^i_{+1}}\rho^i(o^j_{+1} \to o^j_{+2})(\nabla_{\Theta^i}V(\Theta^i,o^{+2}) + \nabla_{\Theta^i}\nabla_{\Theta^j}J(\Theta^j_c,o^j)\Psi(\pi^j_c,a^j,o^j))$

$$\sum_{x^i,x^j \in O}\sum_{k=0}^{\infty}\rho^i(o \to x^i,k)(\Phi(x^i) + \nabla_{\Theta^i}\nabla_{\Theta^j}J_c(\Theta^j_c)\Psi(\pi^j_c,a^j,x^j)) \tag{20}$$

Let $\eta(o) = \sum_{k=0}^{\infty}\rho^i(o^i \to o^{i'},k)$ to clarify the transitions.

$$\sum_o\eta(o)(\Phi(o) + \nabla_{\Theta^i}\nabla_{\Theta^j}J_c(\Theta^j)) \propto \sum_o \frac{\eta(o)}{\sum_o\eta(o)}(\Phi(o) + \nabla_{\Theta^i}\nabla_{\Theta^j}J_c(\Theta^j,o^j)\Psi(\pi^j_c,a^j,o^j) \tag{21}$$

since the normalized distribution is a factor of the sum.

Then let $\sum_s\frac{\eta(o)}{\sum_o\eta(o)} = \sum_{o \in O}d(o)$

$$\sum_{o \in O}d(o)(\sum_{a^i \in A}(\nabla_{\Theta^i}\pi^i(a^i|o^i)Q(o^i,a^i) + \nabla_{\Theta^i}\nabla_{\Theta^j}J_c(\Theta^j,o^j)\Psi(\pi^j_c,a^j,o^j)) \tag{22}$$

$$\sum_{o \in O}d(o)(\sum_{a^i \in A}(\pi^i(a^i|o^i)Q(o^i,a^i)\frac{\nabla_{\Theta^i}\pi^i(a^i|o^i)}{\pi^i(a^i|o^i)} + \nabla_{\Theta^i}\nabla_{\Theta^j}J_c(\Theta^j,o^j)\Psi(\pi^j_c,a^j,o^j)) \tag{23}$$

, the log-derivative trick can pull out the gradient.

$$\sum_{o \in O}d(o)(\sum_{a^i \in A}((a^i|o^i)Q(o^i,a^i)\nabla_{\Theta^i}\log\pi^i(a^i|o^i) + \nabla_{\Theta^i}\nabla_{\Theta^j}J_c(\Theta^j,o^j)\Psi(\pi^j_c,a^j,o^j)) \tag{24}$$

Finally, the full gradient objective from Eq. (5) is constructed

$$\nabla_{\Theta^i}J(\Theta^i) = \mathbb{E}_{\tau \sim \pi^i,\tau \sim \pi^j}[Q(o^i,a^i)\nabla_{\Theta^i}\log\pi^i(a^i|o^i) + \nabla_{\Theta^i}\nabla_{\Theta^j}J(\Theta^j,o^j)\Psi(\pi_{\Theta^j},a^j,o^j))] \; \square$$

## P.2 SELF CORRECTION TERM

Considering Eq. (3) and Eq. (4) the correction term for agent $i$ is equivalent to the expected collective reward value estimate of

$$\mathbb{E}_{\tau \sim \pi^j}[\nabla_{\Theta^j} J_c(\Theta^j)\Psi(\pi_c^j, a^j, o^j)] = \mathbb{E}_{\tau \sim \pi^j}[Q_c(o^j, a^j)] \tag{25}$$

In turn, the collective value estimate is an approximated prediction of the collective reward at any time

$$\mathbb{E}_{\tau \sim \pi^j}[Q_c(o^j, a^j)] \approx \mathbb{E}_{\tau \sim \pi^j, \tau \sim \pi^i}[r^c] \tag{26}$$

.

However the collective reward is also an approximate of the agent $i$'s collective reward values estimate, if they opened their door first, which is again equivalent to the correction term of agent $j$.

$$\mathbb{E}_{\tau \sim \pi^j, \tau \sim \pi^i}[r^c] \approx \mathbb{E}_{\tau \sim \pi^i}[Q_c(o^i, a^i)] = \mathbb{E}_{\tau \sim \pi^i}[\nabla_{\Theta^i} J_c(\Theta^i)\Psi(\pi_c^i, a^i, o^i)] \tag{27}$$

Therefore, in expectation, the correction terms of both agents are equivalent and symmetric. Objective sharing or policy is not necessary,

$$\mathbb{E}_{\tau \sim \pi^j}[\nabla_{\Theta^j} J_c(\Theta^j)\Psi(\pi_c^j, a^j, o^j)] = \mathbb{E}_{\tau \sim \pi^i}[\nabla_{\Theta^i} J_c(\Theta^i)\Psi(\pi_c^i, a^i, o^i)] \; \square \tag{28}$$

Very critically, this equivalence is in *expectation* and therefore is not an instance of a linear calculation or transform but the average value of one agent's correction term is the same as another when in similar context like opening their door first.

### P.3 CORRECTION TERMS DO NOT CONFLICT WITH INDIVIDUAL OBJECTIVES

A corollary to the construction of the correction term is that if there is no collective reward signal (ex. the agent is performing a single agent task), then the correction degenerates to zero.

For the sake of contradiction, assume that the correction term does not become zero when there is a lack of a collective reward signal such that there exists a value $b \neq 0$. Then,

$$b = \frac{\nabla_{\Theta^j} J_c(\Theta^j)}{\mathbb{E}_{\tau \sim \pi^j}[\nabla_{\Theta^j} \log \pi_c^j(a^j|o^j)]} \tag{29}$$

$$\mathbb{E}_{\tau \sim \pi^i}[\nabla_{\Theta^j} \log \pi_c^j(a^j|o^j)]b = \nabla_{\Theta^j} J_c(\Theta^j) \tag{30}$$

$$\mathbb{E}_{\tau \sim \pi^i}[\nabla_{\Theta^j} \log \pi_c^j(a^j|o^j)]b = \nabla_{\Theta^j}(\sum_{t=0}^{T} r_c^j) = \nabla_{\Theta^j}(0) = 0 \tag{31}$$

$$\mathbb{E}_{\tau \sim \pi^i}[\nabla_{\Theta^j} \log \pi_c^j(a^j|o^j)]b = 0 \tag{32}$$

Since $\mathbb{E}_{\tau \sim \pi^i}[\nabla_{\Theta^j} \log \pi_c^j(a^j|o^j)]$ was a denominator, which cannot be zero due to the actions of the collective policy being uniformly distributed (see Eq. (15) there is only one possibility: $b = 0$.

Therefore, $b = 0$ contradicts the claim. $\square$

Intuitively, $b$ is actually equal to $Q_c$ which is obviously zero when there is no collective reward. This result, although quick, shows that an agent can theoretically learn to solve an individual task without conflicting with learned policies for nonstationary coordination behaviours.

