# OpenReview forum: "The challenge of hidden gifts in multi-agent reinforcement learning"
_ICLR.cc/2026/Conference — ICLR 2026 Conference Withdrawn Submission_

### Official Review · Reviewer_TeS5 · 2025-10-20

**Soundness:** 1
**Presentation:** 4
**Contribution:** 2
**Rating:** 2
**Confidence:** 5

**Summary:**

The paper considers the credit assignment problem in multi-agent RL by proposing a grid-world game where agents collect a key to open doors. Opening individual doors gives rewards to corresponding agents, and opening all doors gives a collective reward to all agents. Agents can drop keys so that other agents can collect the key to open their own door. The authors empirically ran a number of prior MARL baselines and found that they did not learn to drop the key. They then derive a corrective term for this game and empirically show that this term improves learning success.

**Strengths:**

- Credit assignment in MARL is an important and difficult problem.
- Proposing new games to crystallize the problem can be a valuable approach.
- A number of MARL algorithms and ablations were run on the proposed game, establishing diverse baselines.
- The paper attempts to give formal explanations, which is nice (but see formal errors, below).
- The paper is well written.

**Weaknesses:**

Unfortunately, the paper contains some important formal errors that invalidate several key claims:

1) In Section 3, the game is formalized as a Dec-POMDP, but the formalization is not correct:

1a: The state is defined as the combined observations of agents. However, observations are local 3x3 tiles, which collectively may not reveal the true state (for example, if all agents are in the same corner, their combined observations may not show the doors and key in the opposite corner).

1b: The reward function in Eq(1) defines individual agent rewards, but Dec-POMDPs assume a single shared reward for all agents. Thus the appropriate game model is the Partially Observable Stochastic Game (see the MARL book by Albrecht et al.). Such games require equilibrium analysis rather than maximizing joint returns of agents. Moreover, the conditions in the definition of (1) are not mutually exclusive, meaning that both conditions may be true at the same time (so the function is ill-defined).

2) The proofs for section 5 contain basic errors. In appendix P1:

2a: The gradient of the collective objective for agent j is written in score-function form and then immediately factorized as

E[∇Θj​​logπ~cj​(aj​∣oj​)]E[Qc​(oj​,aj​)]

in Eq(14). The text justifies this by claiming the agents’ policies are “probabilistically independent.” But independence of policies across agents does not imply independence between the score ∇Θj​​logπcj​(aj​∣oj​) and the value Qc​(oj​,aj​) for the same agent and the same distribution, as Qc depends on aj and oj. Even if one (incorrectly) allowed the factorization, the first factor is the expected score, which equals zero under the sampling distribution: Ea∼π​[∇Θ​logπ(a∣o)]=0. The proof then uses this expectation as a denominator in Eq(15), making this ill-defined due to zero-division.

2b: Circular “proof” that the problematic denominator is non-zero: Appendix P.3 argues by contradiction that if the collective reward is absent, the correction term must be zero. After Eq(30), it is claimed that the denominator cannot be equal to zero since it was a denominator, which is circular. It assumes that the division was valid to begin with, which is not correct.

**Questions:**

Q1: The proposed correction (ignoring the formal errors; see above) seems very specific to the particular game proposed. To what extent could this generalize to other games? I think the paper could be strengthened if broader applicability and usefulness of such terms could be demonstrated.

Q2: Have you tried MARL algorithms that aim to learn collaborative strategies in games such as the ones you proposed? For example, Pareto-AC (https://arxiv.org/pdf/2209.14344) can successfully learn in similar games, such as the "Boulder game" used in their experiments.

Q3: I think the fact that all the MARL algorithms did not learn to drop they key requires a more in-depth analysis of the root causes. I have not checked the code, but I wondered whether there might be a bug where the "key" state is not reset after episodes? In this case, it might be rational for agents to learn to want to keep the key, since it will enable quicker rewards in subsequent episodes? In general, given the POSG setting (as opposed to Dec-POMDP; see above), I think the analysis should focus on what is individually rational for agents rather than collectively rational. It may reveal new angles on the problem analysis.

Q4: Given issue 1b, how did the authors apply MARL algorithms that strictly assume identical rewards between agents, such as VDN and QMIX?

---

> ### Author Response · Authors · 2025-11-20
> **Rebuttal**
>
> ## In Section 3, the game is formalized as a Dec-POMDP, but the formalization is not correct:
> > 1a: The state is defined as the combined observations of agents. However, observations are local 3x3 tiles, which collectively may not reveal the true state (for example, if all agents are in the same corner, their combined observations may not show the doors and key in the opposite corner).
>
> It is true that the egocentric images that the agents receive may not fully reveal the true state of the environment. That is why the Manitokan task is a Dec-POMDP and not a Dec-MDP. Notably, the ability of a single RL agent to use such egocentric images to learn to find a key then open a door is a well established benchmark in temporal credit assignment and part of the Minigrid suite. In section E.8 of the appendix, we tested the single agent key to door task to verify that PG, PPO and DQN architectures have the capacity to learn to find the key then open a door for a reward. All models could learn this task so it reasonably follows that the true state can be found in the Manitokan task, it merely requires exploration of the environment.
>
> Indeed, considering multiple agents in the Manitokan task, which is merely a 5 x 5 grid world, the agents collectively have the ability to gather a sequence of observations that fully specifies the state of the environment. Moreover, their policies include an RNN which has sufficient memory to be able to gather and store sufficient information. Thus, we do not believe that the issue that the reviewer raises here is a valid concern: our task is still in-line with the formalization of a Dec-POMDP, since the observations the agents receive can collectively specify the state.
>
> > 1b: The reward function in Eq(1) defines individual agent rewards, but Dec-POMDPs assume a single shared reward for all agents.
>
> We are not sure why the reviewer makes this assertion about Dec-POMDPs. Dec-POMDPs do not assume there is a single shared reward for all agents (see, e.g. https://arxiv.org/abs/1301.3836). As this seminal paper defined it, for a Dec-POMDP “...expected rewards $R(s, a_1, ... , a_m)$ depend on the actions of all agents”. That is precisely the situation that holds in the Manitokan task. ~~Technically, even the individual rewards depend on the other agents because if the other agents pick up the key and do not drop it, then the individual reward is impossible to obtain~~. We acknowledge that this formalization of the reward is nuanced, and we could have done a better job in clarity. ~~why even the individual rewards depend on the actions of the other agents~~. We will do that in a revised version of the manuscript. However, it is incorrect to say the Manitokan task is not a Dec-POMDP due to its reward function.
> *Note, an incorrect description of the individual reward being dependent on the other agents policy has been crossed out. The individual reward only depends on an agent's own policy and its distance to the key which is randomly initialized.
>
> > Thus the appropriate game model is the Partially Observable Stochastic Game (see the MARL book by Albrecht et al.). Such games require equilibrium analysis rather than maximizing joint returns of agents.
>
> There is certainly the potential for some interesting results through equilibrium analysis, but we do not agree with the reviewer’s claim that the Manitokan task is not a Dec-POMDP. Note that Dec-POMDPs are a sub class of Partially Observable Stochastic Games (POSGs). The Manitokan Task qualifies as a Dec-POMDP because the rewards of the agents do in fact depend on the actions of all of the agents (as noted above). The disagreement here seems to stem from the use of the phrase “shared reward function”, which is the phrase used by the textbook the reviewer cites (and the same citation referenced in the textbook (https://dl.acm.org/doi/10.5555/1597148.1597262). But, the original definition of Dec-POMDPs in Bernstein et al. (2000) does not specify that agents must all receive the same reward from a shared reward function, only that the reward function must depend on the action of all the agents. We will clarify these matters in the manuscript.
>
> > Moreover, the conditions in the definition of (1) are not mutually exclusive, meaning that both conditions may be true at the same time (so the function is ill-defined).
>
> The paper does not state that the conditions of the piecewise reward function need to be mutually exclusive so the function is therefore not ill-defined. We will clarify this to avoid confusion.

---

> ### Author Response · Authors · 2025-11-20
> **Continued Rebuttal**
>
> ## The proofs for section 5 contain basic errors. In appendix P1:
> >2a: The gradient of the collective objective for agent j is written in score-function form and then immediately factorized as E[∇Θj​​logπ~cj​(aj​∣oj​)]E[Qc​(oj​,aj​)] in Eq(14). The text justifies this by claiming the agents’ policies are “probabilistically independent.” But independence of policies across agents does not imply independence between the score ∇Θj​​logπcj​(aj​∣oj​) and the value Qc​(oj​,aj​) for the same agent and the same distribution, as Qc depends on aj and oj.
>
> Thank you for asking about Eq. (14) which is also defined in the main text as Eq (3). We discussed this factorization with Reviewer 5tWK as well. One aspect of the text that was perhaps insufficiently clear is that these equations are conditioned on the agent already having opened their door and dropped the key. Thus, the reason that an agent’s expected collective score-function is independent from its expected collective q-value is that after opening its door and dropping the key, there is no other action that the agent can take to acquire the collective reward. The collective q-value now exclusively depends on the other agent who has to pick up the key and open their door for the collective reward.
>
> We will update the text to make this conditioning on the door being open and key being dropped more explicit so as to avoid similar confusion from readers in the future.
>
> > Even if one (incorrectly) allowed the factorization, the first factor is the expected score, which equals zero under the sampling distribution: Ea∼π​[∇Θ​logπ(a∣o)]=0. The proof then uses this expectation as a denominator in Eq(15), making this ill-defined due to zero-division.
>
> Thank you for highlighting this important step. As mentioned above, there is no action an agent can take to acquire the collective reward after dropping the key since it now depends on the other agent. This critically means that every action that can be sampled by the softmax policy is uniformly likely as any other action to be sampled. Therefore, the collective policy itself is a uniform distribution. As a consequence to the actions being uniformly distributed in the collective policy, the expected gradient of the score function is $\\frac{-1}{|\\mathcal{A}|}$ where $\\mathcal{A}$ is the set of actions (Eq. 16). This is not zero since there are six actions in the Manitokan task. As this is a standard result in statistics, we do not re-derive the expected gradient of the score function of a uniform distribution in the revised manuscript. We will mention this in the main text and proofs to avoid confusion.
>
> We appreciate that Reviewer TeS5 brought this to our attention.
>
> > 2b: Circular “proof” that the problematic denominator is non-zero: Appendix P.3 argues by contradiction that if the collective reward is absent, the correction term must be zero. After Eq. (30), it is claimed that the denominator cannot be equal to zero since it was a denominator, which is circular. It assumes that the division was valid to begin with, which is not correct.
>
> Since the denominator can not be zero (see above), our proof by contradiction is not circular but direct. We will mention that the denominator can not be zero due to the independence of the collective score-function and collective q-value in expectation leads to the consequential uniform likelihood of actions in the collective policy in this section of the proof. This is specific to the agent that opens their door and drops the key.
>
> ## Questions:
> > Q1: The proposed correction (ignoring the formal errors; see above) seems very specific to the particular game proposed. To what extent could this generalize to other games? I think the paper could be strengthened if broader applicability and usefulness of such terms could be demonstrated.
>
> In the related works, we noticed that almost all MARL environments were framed as common games that leveraged shared rewards or shared information between agents to solve the tasks. These environments are meant to test if reinforcement learning agents can coordinate to solve a single goal or objective. In the Manitokan task, we wanted to isolate and test something more nuanced, that is, “hidden gifts” where the critical actions are unobserved to the other agents in a collective task that requires a scarce resource used in individual tasks. Therefore, the correction terms can not currently generalize to common games since it corrects for a different problem. We expect if more fully decentralized tasks are developed with individual and collective objectives, the correction terms can be utilized to improve performance and stability.

---

> > ### Author Response · Authors · 2025-11-20
> > **Final Rebuttal**
> >
> > > Q2: Have you tried MARL algorithms that aim to learn collaborative strategies in games such as the ones you proposed? For example, Pareto-AC (https://arxiv.org/pdf/2209.14344) can successfully learn in similar games, such as the "Boulder game" used in their experiments.
> >
> > All algorithms we evaluated have previously been used to learn collaborative strategies in fully or mixed cooperative games. COMA, in particular, uses a baseline in its advantage estimation that considers alternative joint actions across agents. This baseline is closely related to Pareto-AC, which argmaxes over the actions of the other agents to maximize the Q-value for a given agent. Since COMA already considers those alternative actions between agents that would have improved its value estimate and return, we view Pareto-AC as too similar in spirit to COMA, and therefore consider this design avenue to have already been explored.
> >
> > Additionally, the Boulder Game is not directly comparable to the Manitokan task. In the Boulder Game, agents receive a negative reward for pushing the boulder alone to inhibit miscoordination, whereas the Manitokan task, as noted in the introduction, does not include any negative rewards. While both environments involve a collective goal and allow for critical actions to be unobserved, the Boulder Game does not include any notion of gifting between agents. Thus, the Boulder Game differs significantly from the Manitokan task.
> >
> > One could, in principle, modify the Manitokan task to resemble the Boulder Game. For example, by introducing two keys and a single door that requires both agents to open it simultaneously. However, such a modification fundamentally changes the nature of the task and removes the core challenge of hidden gifts.
> >
> > > Q3: I think the fact that all the MARL algorithms did not learn to drop they key requires a more in-depth analysis of the root causes. I have not checked the code, but I wondered whether there might be a bug where the "key" state is not reset after episodes?
> >
> > A more in-depth analysis of both the representations learned by deep MARL policies and the effects of the mechanisms used to induce cooperation is an interesting direction for future work, but lies beyond the scope of the present paper. We have also clarified the novelty of our contribution in our response to Reviewer 5tWK.
> >
> > The speculation about a potential bug in the “key” state is incorrect. We believe this misunderstanding stems from some MARL environments maintaining environmental states or objects across episodes and policy updates such as Melting Pot. In contrast, the Manitokan task is implemented using Minigrid, which, by design, resets key states at the end of every episode (and thus at every policy update). We did not alter this feature of Minigrid. Moreover, if such a bug existed and the key was not being reset, we would expect the reward plots to show only one agent acquiring the reward. Instead, all reward plots clearly show both agents receiving rewards.
> >
> > > In this case, it might be rational for agents to learn to want to keep the key, since it will enable quicker rewards in subsequent episodes? In general, given the POSG setting (as opposed to Dec-POMDP; see above), I think the analysis should focus on what is individually rational for agents rather than collectively rational. It may reveal new angles on the problem analysis.
> >
> > Earlier, we explained that the Manitokan task satisfies the definition of a Dec-POMDP, with a nuance in how the individual rewards are necessary to obtain the collective reward. Throughout the main text, all equations are actually written from the perspective of an individual agent indexed with $i$ or $j$ so, we do in fact analyze what is individually rational for a single agent in the Manitokan task. Because there are no negative rewards, the collective reward is strictly larger than the individual reward and can only be obtained once the individual reward has been collected; in this setting, what is individually rational for a reward maximizing agent is also collectively rational. We then leveraged this in our formal analysis on how individual agents, through their policies, influence the collective reward to derive a decentralized correction term which is also individualistic.
> >
> > > Q4: Given issue 1b, how did the authors apply MARL algorithms that strictly assume identical rewards between agents, such as VDN and QMIX?
> >
> > VDN and QMIX do not assume identical rewards themselves per se but rather their original papers only tested them on environments with identical rewards (https://arxiv.org/abs/1706.05296 and https://arxiv.org/abs/1803.11485). It was reasonable to assume that the value decomposition mechanisms could align the agents' values to the larger collective reward in the Manitokan task. We found that value decomposition does not solve the Manitokan task but this empirical finding is still important for the MARL community to consider future research directions.

---

### Official Review · Reviewer_5tWK · 2025-10-29

**Soundness:** 1
**Presentation:** 3
**Contribution:** 1
**Rating:** 0
**Confidence:** 4

**Summary:**

The paper proposes a new learning task that focuses specifically on the “hidden” collaboration behaviours (dubbed “hidden gifts”) in the multi-agent reinforcement learning. It evaluates several existing MARL algorithms on this task and shows that all of them failed to learn meaningful policies. The paper then proposes a new policy gradient theorem that has an extra correction term and has been shown to work well on this learning task.

**Strengths:**

The paper proposes an interesting learning task for multi-agent reinforcement learning (MARL): Manitokan task. This task emphasizes specifically on the collective behaviour among agents (called the hidden gifts). The paper conducted a relatively comprehensive evaluation of mainstream MARL methods and reported their performance on this task.

The paper is well written and easy to read. All results are presented in a clear and understandable way.

**Weaknesses:**

The formal analysis is not rigorous and the contributions are unclear
### Lacks of rigour
1. The expectations in all equations are not specified or clearly defined, which makes some equations suspiciously wrong (or with factual errors). For example, Eq. (2) is inconsistent with the definition of reward in Eq. (1), the latter of which specifies two cases but the former just sums up these two cases; Eq. (3) appears to be incorrect if the expectation involves the summation over action space; in Theorem 1, $\Psi$ is defined as the expected value of reciprocal, which is a gradient with respect to $\Theta$. What does it mean for a reciprocal of a vector?

2. Most results relies on the (possibly incorrect) configurations of the test environments. Specifically, the Dec-POMDP might not be correctly defined in the first place. At the least the reward function in Eq. (1) shows that $\mathcal{R}$ is not only a function of $o_t$ and $a_t$ but also a function of the status of all doors. Second, the results in Section 4.1 and Section 4.2 essentially verify that the configurations of the environment were not correct initially: that the Dec-POMDP formulation couldn’t be used to maximize desired rewards, which was later confirmed by the results in Section 4.3. However, Section 4.3 only considers a very simple improvement: adding the last action as a one-hot vector (why not using the full action history?).

### Unclear contributions
1. There is no in-depth analysis into any MARL algorithms or their poor performance Among all these results, only the success rate and cumulative rewards are reported, which can hardly be used to analyse why these algorithms failed to learn. For example, if the last action is added as extra info, the paper should at least verify whether such last action is really being used by the policy learning with a GRU unit (whether more previous actions are needed?) Also, the reward for each agent is dependent on the order when the agent picks up the key: if the first one, then reward is small; if the last one, the reward is then increased. This would affect the value estimation as well: the paper should then check if such order info has been provided in value approximation?

2. The empirical results provided in Figure 5 are obtained from training with significantly more episodes than that in Figure 2, 3, and 4. This makes the results in Figure 2, 3, and 4 less convincing since the early stopping of the training can lead to completely different conclusions. For example, if the training terminates at 7500 episodes in Figure 5, then success rate and cumulative reward would be very similar to those reported in other figures.

**Questions:**

1. What’s the definition of expectation in Eq. (2), (3), (4) and (5)?
2. Why the reward in Eq.(2) is the sum of two terms? (the reward was previously defined as the piecewise function)
3. the $\Psi$ in Theorem 1 is not correctly defined: the reciprocal of vector makes no sense.

---

> ### Author Response · Authors · 2025-11-20
> **Rebuttal**
>
> ## Lacks of rigour
> > The expectations in all equations are not specified or clearly defined, which makes some equations suspiciously wrong (or with factual errors).
>
>  We recognize that we were not explicit with the expectations, though we do not agree that there were errors in the analysis as a result. To be fully clear now, all expectations are expectations over trajectories sampled from the policy of an agent given a randomly initialized state. We will modify the document to include these terms as well as summations to improve clarity of the work. Notably, the expectations are derived in the appendix with the derivation of the policy gradient theorem.
>
> > For example, Eq. (2) is inconsistent with the definition of reward in Eq. (1), the latter of which specifies two cases but the former just sums up these two cases;
>
> Thank you for finding the typo in Eq. (2), which should use the scalarized reward function, $\\hat{R}$, not the vectorized reward from Eq. (1). To motivate this, we note that it is not standard for reinforcement learning agents to learn from vectors of rewards from a piecewise function and a common technique in multi-objective literature is to scalarize the rewards by adding them in the sampled trajectory. This technique was discussed in the related works section for Multi-Objective Reinforcement Learning with referenced papers. We defined the scalarized reward function directly under Eq. (1) but missed using $\\hat{R}$ in Eq. (2), we will fix that. Note also that the form of $\\hat{R}$ used in Eq. (2) is based on the fact that the empirically sampled rewards will naturally incorporate the weighting based on the other agents’ policy. We will be sure to note the use of the scalarization technique and empirical weighting in the text around Eq. (2).
>
> > Eq. (3) appears to be incorrect if the expectation involves the summation over action space;
>
> Thank you for asking about Eq. (3) which is the critical step used to derive the correction terms. We recognize that this step caused confusion amongst several reviewers, which indicates a lack of clarity that we intend on addressing. Briefly, the factorization of  $\\mathbb{E} [\\log(\\pi_c(a|o))]$ and $\\mathbb{E}[Q_c(o,a)]$ holds only when the agent has already opened their door and dropped their key, which was noted in the text. In this case, nothing the agent does will affect the collective reward – once an agent has opened their door and dropped their key, only the actions of the other agent can determine the Q-value for the current observation. Hence, $\\mathbb{E}[Q_c(o,a)]$ is actually independent of the policy gradient, and Eq. (3) holds. We will update the text to make this conditioning on the door being open and key being dropped more explicit so as to avoid similar confusion from readers in the future.
>
> > in Theorem 1, s defined as the expected value of reciprocal, which is a gradient with respect to . What does it mean for a reciprocal of a vector?
>
> Thank you for asking about Eq. (4) which is the other critical step used to derive the correction terms (and also another source of confusion for some of the reviewers). By the reciprocal of a vector, we just mean element-wise reciprocals, i.e. each element of the reciprocal vector is just 1 over the original vector element. We will update the text to clarify this.
>
> > Most results rely on the (possibly incorrect) configurations of the test environments. Specifically, the Dec-POMDP might not be correctly defined in the first place.
>
> We are confident that the environment is configured correctly, and that the Dec-POMDP is well formed. But, it was built to differ from past environments mentioned in the related works section, which makes it unique, and that can lead to confusion. Below we explain why the Dec-POMDP is well-configured (see also the replies to Reviewer TeS5 with additional discussion of the Dec-POMDP).

---

> > ### Author Response · Authors · 2025-11-20
> > **Continued Rebuttal**
> >
> > > At the least the reward function in Eq. (1) shows that is not only a function of  and but also a function of the status of all doors. Second, the results in Section 4.1 and Section 4.2 essentially verify that the configurations of the environment were not correct initially: that the Dec-POMDP formulation couldn’t be used to maximize desired rewards, which was later confirmed by the results in Section 4.3. However, Section 4.3 only considers a very simple improvement: adding the last action as a one-hot vector (why not using the full action history?).
> >
> > Respectfully, we do not agree with the reviewer that our task was ill-defined, either in the formal analysis, or the experiments. For example, in response to the reviewer’s specific note regarding the doors, the opening status of the doors is already embedded into the observation space through the door disappearing when opened. Given that the agents have some memory (because they are all implemented with RNNs) it is possible, in principle, for the agents to learn the task purely from the observations they received – for example, an agent could learn that if they no longer see their door they should drop the key. This is why Eq. (1) is perfectly well formulated: the observations o_t contain the necessary information regarding the status of the doors. However, what we can say is that in this Dec-POMDP, the status of the doors is just partially observable, which is part of why this is a challenging task.
> >
> > Our goal in this study was to only add as much information as necessary to make the task solvable. Obviously, if we made this task a standard MDP, e.g. by providing the agents with complete state information as inputs, then we would expect all of the models to be able to solve the task. Based on the reviewer’s question, though, we have also run additional experiments where we provide more action history in the observation. We find that the results of adding more actions are mixed: it can both help and hinder, though we do find it can only help when the temporal order of the actions is preserved (see the new figure in the Appendix section E.8). These results indicate that a single previous action provides a great deal of the necessary information to solve the task, though more history can help, which supports the conclusion that the task is not ill-specified and agents can use the provided information to maximize rewards, it is simply challenging.
> >
> > This brings us to the central aspect of our contribution, which is the highly decentralized nature of our task. In our task, not only are the important state variables partially observable, but the state is uniquely determined from the observations of all the agents (which, collectively, provide information about the status of all the doors and keys). Part of the interesting aspect of “hidden gifts” is precisely that the information necessary for credit assignment is decentralized, and some key aspects of information reside with the other agent.
> >
> > ## Unclear contributions
> > > There is no in-depth analysis into any MARL algorithms or their poor performance
> >
> > To specify the contributions, we proposed a novel structural credit assignment problem to test if reinforcement learning agents are able to learn to let go of a scarce item they used to receive a reward so the other agent and themselves can receive a reward. This is a novel Dec-POMDP configuration as mentioned above where the collective reward is causally conditioned on the acquiring the individual rewards. We then tested 9 different state of the art MARL algorithms designed for fully cooperative games. None of which were able to learn the task above at random. We then hypothesized and tested what decentralized information could spur cooperation. Only the last action in the current observation led to success which we then leveraged a novel decentralized learning awareness term to correct for bias in the policy updates.
> >
> > A deeper analysis into both the representations being learned in the policies of deep MARL agents and the effects of the mechanisms meant to induce cooperation is a very interesting future direction, but we consider it to be outside the scope of this work.
> >
> > > Among all these results, only the success rate and cumulative rewards are reported,
> >
> > Further experiments are reported in the appendix, and they are referenced throughout the paper.
> >
> > > which can hardly be used to analyse why these algorithms failed to learn. For example, if the last action is added as extra info, the paper should at least verify whether such last action is really being used by the policy learning with a GRU unit (whether more previous actions are needed?)
> >
> > See above, the new last action experiments address this in Appendix section E. 10.

---

> > > ### Author Response · Authors · 2025-11-20
> > > **Final Rebuttal**
> > >
> > > > Also, the reward for each agent is dependent on the order when the agent picks up the key: if the first one, then reward is small; if the last one, the reward is then increased. This would affect the value estimation as well: the paper should then check if such order info has been provided in value approximation?
> > >
> > > Order information was not provided to value approximation but, the key-door status experiments in Figure 3 did include order information implicitly since the signal for having opened the door would imply that the agent picked up their key first. This information did not improve the performance of value-based models we tested (VDN and QMIX). Additionally, order information could also be inferred with last actions which also did not improve the performance of value-based models we tested.
> > >
> > > > The empirical results provided in Figure 5 are obtained from training with significantly more episodes than that in Figure 2, 3, and 4. This makes the results in Figure 2, 3, and 4 less convincing since the early stopping of the training can lead to completely different conclusions. For example, if the training terminates at 7500 episodes in Figure 5, then success rate and cumulative reward would be very similar to those reported in other figures.
> > >
> > > Respectfully, there was no early stopping of any experiments. The runs plotted in Figure 2, 3, and 4 were truncated at 10000 episodes to accommodate a fair comparison between different algorithms due to architectural structure that slows down overall training. Particularly,  PPO based models (MAPPO, IPPO, SAF) use an iterative bootstrapping procedure to calculate their advantages while PG does not. Some value mixer models (VDN, QMIX, QTRAN) required large batch sizes to guarantee a collective reward signal and some have deeper models to calculate their global value function approximations (QMIX, QTRAN, MAVEN). Figure 5 only includes extended runs due to the improved performance in Figure 4 where other models still did not perform.
> > >
> > > We have some experiments where models ran further than 10000 episodes but they did not exhibit a change in performance.
> > >
> > > ## Questions:
> > > > What’s the definition of expectation in Eq. (2), (3), (4) and (5)?
> > >
> > > All expectations are expectations over trajectories sampled from the policy of an agent given a randomly initialized state.
> > >
> > > > Why the reward in Eq.(2) is the sum of two terms? (the reward was previously defined as the piecewise function)
> > >
> > > The reward in Eq. (2) is the scalarized reward function, $\hat{R}$, not the piecewise reward
> > > from Eq. (1). This is a technique from multi-objective reinforcement learning discussed in
> > > the related works section.
> > >
> > > > The $\Psi$ in Theorem 1 is not correctly defined: the reciprocal of vector makes no sense.
> > >
> > > This is an element wise reciprocal of a vector, as mentioned above.

---

### Official Review · Reviewer_turr · 2025-10-29

**Soundness:** 2
**Presentation:** 2
**Contribution:** 2
**Rating:** 4
**Confidence:** 3

**Summary:**

The paper investigates why state-of-the-art multi-agent reinforcement learning (MARL) algorithms struggle on certain cooperative tasks, focusing on a simple yet challenging benchmark called the Manitokan task. The authors argue that this task reveals failures in credit assignment—the ability of agents to attribute collective outcomes to individual actions. To study this, they introduce a minimal domain that supports both mathematical and empirical analysis. They observe that most MARL algorithms, including MAPPO, COMA, and QMIX, fail to make learning progress on Manitokan, whereas single-agent baselines and decentralized agents with action history perform better. The authors also derive a self-correction term motivated by their theoretical analysis of policy gradients in this environment, which is intended to address learning inefficiencies observed in their experiments.

**Strengths:**

- The problem is clearly motivated and addresses an important gap in our understanding of MARL failure modes.
- Several empirical findings are interesting, including that most algorithms tested struggle to make learning progress on Manitokan, and that adding action history helps decentralized agents but not MARL agents.
- I appreciate the approach of proposing a minimal, analytically tractable toy problem and performing both theoretical and empirical analysis within that setting.

**Weaknesses:**

- The claim that MARL algorithms struggle on Manitokan primarily due to credit assignment issues is not convincingly substantiated. The paper does not clearly define “credit assignment,” offering only intuitive examples. Nor does it show how such difficulties mathematically impact the gradients or losses of the algorithms studied. Alternative explanations—such as biased gradients in MAPPO or the presence of local minima—were not ruled out. No empirical probes were conducted to directly verify that credit assignment was the key bottleneck.
There is no clear evidence that “hidden gift” type problems, in general, pose unique challenges for MARL algorithms beyond the specific Manitokan setup.
- The evidence that state-of-the-art MARL algorithms cannot solve Manitokan is limited. In Figure 4, these algorithms experience a decrease in reward before 8K episodes, but Policy Gradient—tested out to 24K episodes—shows the same initial drop before eventually achieving a high success rate. It seems plausible that MAPPO, COMA, or QMIX might have reached similar or higher success rates if trained for longer.
- The significance of the self-correction term is unclear. It appears to rely on assumptions specific to Manitokan, leaving open whether it generalizes to other domains. Moreover, it does not appear to substantially improve success rates. While it may slightly reduce return variance, the paper provides no confidence intervals (e.g., in Fig. 5c) to assess whether this effect is statistically meaningful.
- The plots use colors that are difficult to distinguish, especially in Figure 5, where the lines for Vanilla and Self-Correction are nearly indistinguishable.
- The mathematical analysis lacks intuitive interpretation. For example, what is the meaning of Equations 3–5, and how do they concretely relate to the credit assignment problem? What is the intuition behind the correction term, and why should it improve learning?
- The theoretical section also omits key definitions and justifications. For instance, what precisely is meant by a sub-policy? This concept does not seem to be rigorously defined, either in the paper or in the cited work by Sutton. Several assumptions in the derivations are also stated without explanation or justification, making it difficult to verify correctness.

**Questions:**

- Regarding the correction term: is the claim that the gradients computed by standard policy gradient are biased or otherwise incorrect? What precisely is being “corrected”?
- How do you formally define credit assignment in your framework?
- What is a sub-policy, and how does it differ from an agent’s standard policy?
- What is the conceptual significance of Equation 3?
- In your derivation, how are the expectations over the log-policy term and the Q-function separated? It may help to explicitly state what each expectation is taken with respect to.
- In Equation 16, is there a missing equals sign or continuation marker? It’s unclear whether this is part of the preceding expression.

---

> ### Author Response · Authors · 2025-11-25
> **Rebuttal**
>
> ## Weaknesses:
> > The claim that MARL algorithms struggle on the Manitokan task primarily due to credit assignment issues is not convincingly substantiated. The paper does not clearly define “credit assignment,” offering only intuitive examples. Nor does it show how such difficulties mathematically impact the gradients or losses of the algorithms studied. Alternative explanations—such as biased gradients in MAPPO or the presence of local minima—were not ruled out. No empirical probes were conducted to directly verify that credit assignment was the key bottleneck. There is no clear evidence that “hidden gift” type problems, in general, pose unique challenges for MARL algorithms beyond the specific Manitokan setup.
>
> We think the concern here comes from a mismatch in terminology, so we will make our usage more explicit in the paper. By credit assignment we mean structural credit assignment in MARL: the problem of attributing joint return to an agent’s own observations and actions when reward is delayed and depends on both agents’ policies. The Manitokan task instantiates this directly through its reward structure, which combines an individual reward with a larger collective reward that is only realized if both agents succeed. Because the collective reward is conditioned on the interaction of the two policies, each agent must assign credit not only to finding the key and opening its own door, but also to the altruistic action of leaving the key for the other agent. Under the task reward, the correct attribution places greater credit on the altruistic sequence that enables the larger collective payoff. This is the structural credit assignment setting we referenced in the introduction and the Manitokan task section, and it aligns with MARL formulations of structural credit assignment in referenced work (see https://ntrs.nasa.gov/citations/20040068179 and https://proceedings.neurips.cc/paper/2021/hash/fe1f9c70bdf347497e1a01b6c486bdb9-Abstract.html ).
>
> Regarding the suggested alternatives, we do not view “biased gradients” or “local minima” as separate explanations in this setting, because both are manifestations of misattribution under this reward structure. If an agent’s gradient is biased toward immediate individual return, that bias arises from failing to assign sufficient credit to the delayed collective return enabled by gifting. Likewise, the greedy local optimum corresponds to assigning credit to individually rewarded key-use while under-crediting the altruistic action that is necessary for the collective outcome. These phenomena  are consequences of the same structural credit assignment problem.
>
> The significance of “hidden gifts” should be interpreted relative to the kinds of MARL environments that dominate the literature and that we summarized in the related works section. The majority of those benchmarks are common-reward or common-goal games: they test whether agents can coordinate to achieve a shared objective where rewards are aligned and directly shared. A recurring feature of successful approaches on these problems is explicit or implicit information sharing between agents, typically via centralized critics or centralized-training–decentralized-execution schemes.
>
> The Manitokan task is different in a distinct way. It evaluates whether agents can learn to altruistically relinquish an item that is immediately useful for their own individual reward, so that all agents can later obtain a larger collective reward that is conditioned on both agents’ policies. Empirically, every SOTA MARL method we tested performs strongly on the standard common-reward games, yet fails to solve the Manitokan task, even when we provide extra decentralized state information. This contrast is the concrete basis for our claim that hidden-gift structure exposes a MARL failure mode not covered by existing benchmarks. This hidden-gift credit structure is a meaningful and currently under-tested failure mode, consistent with the reviewer’s own comment of our paper’s strengths that the Manitokan task “... addresses an important gap in our understanding of MARL failure modes” .

---

> > ### Author Response · Authors · 2025-11-25
> > **Continued Rebuttal**
> >
> > > The evidence that state-of-the-art MARL algorithms cannot solve Manitokan is limited. In Figure 4, these algorithms experience a decrease in reward before 8K episodes, but Policy Gradient—tested out to 24K episodes—shows the same initial drop before eventually achieving a high success rate. It seems plausible that MAPPO, COMA, or QMIX might have reached similar or higher success rates if trained for longer.
> >
> > This was brought up by Reviewer 5tWK as well but to be succinct, many MARL models like MAPPO and QMIX do not permit the same amount of training time as the PG models due the architectural design and the academic compute restraints we are under. Particularly, the PPO’s bootstrapping advantage estimation and QMIX’s large batch sizes for training. We could have, alternatively, never shown the PG model experiments past 10000 episodes but this would obscure some important results related to the longer term stability of our self-correction term solution. We do have some data we ran as a sanity check for some models that extend further than 10000 episodes but there was no improvement in performance nor change in behaviour.
> >
> > > The significance of the self-correction term is unclear. It appears to rely on assumptions specific to Manitokan, leaving open whether it generalizes to other domains. Moreover, it does not appear to substantially improve success rates. While it may slightly reduce return variance, the paper provides no confidence intervals (e.g., in Fig. 5c) to assess whether this effect is statistically meaningful.
> >
> > The correction and self-correction terms do leverage the reward function structure but are  derived from Learning with Opponent Learning Awareness (LOLA) to handle the kind of  dynamic learning signal present: the collective reward is conditioned on both agents’ policies and therefore shifts as either agent updates. As mentioned above, most MARL environments are common goal games where all agents are directly aligned by shared rewards, and SOTA performance is typically achieved via information sharing, most often through centralized critics or CTDE. The Manitokan task deliberately departs from that regime by requiring altruistic transfer: an agent must give up an item that is immediately useful for its own individually rewarded progress so that both agents can later secure a later, larger collective payoff.
> >
> > The significance of the self-correction term is that it improves learning in a setting with necessary a individual and a collective reward, i.e., a multi-objective cooperative problem. Where only the collective payoff depends on other agents’ changing policies, so treating the collective reward signal as stationary leads to misaligned updates. Self correction brings in a fully decentralized learning-awareness adjustment, making each agent account for how their policy updates affect the future collective reward. In this sense, self-correction extends the learning-awareness paradigm (e.g., LOLA) as a solution to a new multi-agent “hidden gift” problem.
> >
> > The success-rate gains from improved stability are not easy to read from episode-wise trajectories, because collective success is sparse and delayed and the curves are correspondingly noisy. After 5tWK’s comment,  we included the box plot of global median collective success across all policy-gradient models: in that aggregate view, the self-correction term shows a clear improvement in collective success relative to the same baselines without it. This can be found in appendix experiment E.12.  In Fig 5c, confidence intervals like we had in previous plots are inappropriate here since this plot communicates average variance across simulations over episodes. To make the stability effect visually legible, we additionally shaded the area under the variance curves, so the distance to zero variance can be read directly.
> >
> > > The plots use colors that are difficult to distinguish, especially in Figure 5, where the lines for Vanilla and Self-Correction are nearly indistinguishable.
> >
> > We thank the Reviewer for pointing out the similarity of colours in Figure 5. We changed them to be more inclusive and dissimilar in the new revised paper.

---

> > > ### Author Response · Authors · 2025-11-25
> > > **Continued Rebuttal**
> > >
> > > > The mathematical analysis lacks intuitive interpretation. For example, what is the meaning of Equations 3–5, and how do they concretely relate to the credit assignment problem? What is the intuition behind the correction term, and why should it improve learning?
> > >
> > > The balance of intuition and rigour is very important when communicating technical concepts. Notably Reviewer 5tWK felt we were too intuitive and lacked rigour. Nonetheless, intuition is critical for everyone to understand.
> > > Equations (3)–(5) are the core derivation of the correction term, whose purpose is to account for how changes in another agent’s policy alter the collective reward across episodes. They start from the scalarized reward under Eq (1), which decomposes into an individual component plus a collective component. Eq. (3) isolates the collective sub-objective and is then differentiated as a policy-gradient. Its meaning is: “what direction would an agent update or what actions would an agent repeat if it were optimizing only the collective reward?” This isolates the part of learning where credit must be assigned to behaviors that matter only for joint collective return.
> > >
> > > In the Manitokan task, the collective reward has a particular causal structure: given the task logic of the reward function, there is no action by the first agent that can unilaterally increase the collective return. Only the second agent can do so. This is why, in expectation, the score-function term for the first agent is statistically independent of its collective Q-value estimate, and why the first agent’s collective policy is uniform in expectation under the collective sub-objective. Intuitively, Eq. (3) is a consequence to this: the collective payoff cannot be attributed to a locally optimal action of the first agent, so naïve gradients provide no directional credit for the altruistic behavior that enables the joint collective outcome.
> > >
> > > Eq. (4) then uses the correction term as a surrogate for the change in collective reward inside the look-ahead step of the policy-gradient theorem. The element-wise division exploits the fact that the collective policy provides is uniform in expectation. Intuitively, Eq. (4) is how “the other agent’s learning changes my collective return” but in a differential term.
> > >
> > > Eq. (5) is the final optimization target. Its meaning is: “update my policy to maximize expected cumulative reward while correcting for the fact that the collective reward is non-stationary because the other agent is learning.” This directly links to credit assignment because, under Eq. (5), an agent’s gradient no longer assigns credit solely to what is immediately reinforced under a fixed partner. Instead, the agent assigns credit in a way that anticipates how the current policies’ behavior shapes the nexts’ collective payoff through the other agent’s update (or through its own update in the self-correction variant). The intuition for why this should improve learning is exactly the LOLA learning awareness idea: in environments where joint reward depends on learning agents, anticipating and correcting for another's learning yields a more accurate credit signal for their actions.

---

> > > > ### Author Response · Authors · 2025-11-25
> > > > **Final Rebuttal**
> > > >
> > > > > The theoretical section also omits key definitions and justifications. For instance, what precisely is meant by a sub-policy? This concept does not seem to be rigorously defined, either in the paper or in the cited work by Sutton. Several assumptions in the derivations are also stated without explanation or justification, making it difficult to verify correctness.
> > > >
> > > > We acknowledge that we treated “sub-policy” as more standard terminology than it might be. In our paper, a sub-policy is actually rigorously defined at the point where we decompose the scalarized reward and derive the collective reward sub-objective: it is the policy obtained when optimizing only the collective reward component, in contrast to the full policy that optimizes the full maximize reward objective. We cited Sutton’s work since it does mention sub policies, albeit in an options framework. We agree that the semantic link is not be entirely clear so we will include references to hierarchical learning (https://sferics.idsia.ch/pub/juergen/grindelwald2004bakker.pdf and https://proceedings.mlr.press/v70/andreas17a.html) so that past descriptions of sub-policies can better understood by the reader.
> > > >
> > > > On assumptions and justifications, the derivations follow the standard policy gradient derivation, including the steady-state distribution steps, and rely on the same baseline conditions used in that literature. For the non-standard parts, where we introduce the correction and self-correction terms or prove their properties, we stated the mathematical step being used at each line and referenced the specific property that motivates it. If there are specific lines Reviewer turr believes lacks justification, we can expand that step directly, but as written the derivation is intended to be verifiable given a standard reinforcement learning background.
> > > >
> > > > ## Questions:
> > > > > Regarding the correction term: is the claim that the gradients computed by standard policy gradient are biased or otherwise incorrect? What precisely is being “corrected”?
> > > >
> > > > The gradients computed by the standard policy gradient term are biased since they do not account for how the policies of agents change and affect the collective reward. We also elaborated on this above.
> > > >
> > > > > How do you formally define credit assignment in your framework?
> > > >
> > > > The credit assignment problem that we are studying (i.e hidden gifts) is formally defined in the construction of the Manitokan task where agents have to successfully learn to associate an individual reward with the actions of finding a key and opening their door then leaving the key for another agent. We described this credit assignment problem as structural.
> > > >
> > > > > What is a sub-policy, and how does it differ from an agent’s standard policy?
> > > >
> > > > The sub-policy is defined in the sub-objective of maximizing only the collective reward. In this problem we are able to further derive a correction term for the change in the collective reward through the sub-objective. We needed to break apart the standard policy into sub-policies to analyze the problem at hand.
> > > >
> > > > > What is the conceptual significance of Equation 3?
> > > >
> > > > This is the first critical step in deriving the correction term that leverages how the Q-value estimate for the collective reward is dependent on the other agent’s policy
> > > >
> > > > > In your derivation, how are the expectations over the log-policy term and the Q-function separated? It may help to explicitly state what each expectation is taken with respect to.
> > > >
> > > > Each expectation is taken with respect to a trajectory sampled by an agent's policy with a randomized initial state. We added this to in the new revision. We described the separation step detail above and with other reviewers.
> > > >
> > > > > In Equation 16, is there a missing equals sign or continuation marker? It’s unclear whether this is part of the preceding expression.
> > > >
> > > > No, there is no missing equals sign or continuation marker. Equation 16 in the original submission is intended to define $\Psi$ via the correction term and to state that this term is equivalent to the collective Q-value estimate used in the proof. Stylistically, we presented each logical step as a new displayed equation rather than writing one long multi-line chain or multiple lines with “$=$”, because that lets us reference and explain each step in the surrounding text. Writing the same argument as a single continuation (e.g., $\cdots = \cdots = \cdots$) would be an equally valid proof format, but it would compress distinct sub-steps and make it harder to attach specific verbal justification to each step.

---

### Official Review · Reviewer_ZmqS · 2025-10-31

**Soundness:** 1
**Presentation:** 2
**Contribution:** 1
**Rating:** 2
**Confidence:** 4

**Summary:**

This paper introduces the "Manitokan task," a simple cooperative grid-world environment designed to study the problem of "hidden gifts." In this task, agents must share a single key to open their individual doors, but the act of one agent dropping the key (the "gift") for another is not directly observable. The authors show that several state-of-the-art MARL algorithms (including VDN, QMIX, COMA, and MAPPO) fail to solve this task, often collapsing to policies with zero cooperative behavior. The paper finds that a simple decentralized Policy Gradient (PG) agent, when provided with its own action history, can learn the task, albeit with high variance. The authors then derive a "Self Correction" term, inspired by learning-aware approaches, which is shown to reduce this variance and improve convergence to the cooperative solution.

**Strengths:**

- **Problem Formulation**: The paper identifies an interesting and potentially challenging problem in MARL: credit assignment for cooperative actions that are not immediately or directly observable by the beneficiaries.

- **Empirical Analysis of Baseline Failures**: The demonstration that a wide range of standard MARL baselines fails on this seemingly simple task is a valuable finding. The analysis of this failure (e.g., collapse of key-dropping behavior) provides a clear motivation for the work.

- **Proposed Solution**: The derived "Self Correction" term, while simple, is empirically shown to be effective. The results in Figure 5 demonstrate that this term reduces variance and improves the collective success rate compared to vanilla PG and a max-entropy baseline. Although this experiment by itself is not robust to make the claim.

**Weaknesses:**

This paper, while presenting an interesting idea, suffers from several major weaknesses in its problem formulation, experimental setup, and theoretical grounding.

- **Unclear Definition of "Hidden Gift"**: The central premise of the "hidden gift" is unclearly defined and potentially flawed. The paper states "the act of dropping the key is not actually observable" (line 189). However, it is not easy to know the experimental setup if:
  - (a) Only the action $a_{\text{drop}}$ is hidden from other agents' observations $o^j$, but the resulting state change (i.e., the key appearing on the ground at $s'$) is observable?
  - (b) The resulting state change is also hidden (e.g., the key is invisible to agent $j$ even after agent $j$ picks it up)?

If (a), then this is not a new "hidden gift" problem but a standard MARL problem of non-stationarity and multi-step temporal credit assignment, where the state change is induced by another agent's policy. If (b), this is an extremely strong partial-observability assumption that needs much clearer justification.

- **Extreme and Unjustified Partial Observability**: The initial experiments (Sec 4.1) appear to be conducted under extreme partial observability, where "agents did not have an explicit signal for their door being opened or that they are holding the key" (lines 268-270). This lack of basic proprioceptive state (knowing if one is holding the key or has completed one's own task) makes the problem artificially difficult and is a highly unusual assumption for MARL environments. This should be made far clearer and be justified.

- **Missing Key Baselines and Related Work**: The paper's primary claim is that existing MARL algorithms fail at this credit assignment problem. However, it omits a large and highly relevant body of work specifically designed to address non-stationarity and credit assignment in CTDE settings.

  - The paper notes issues like "conflicting gradient updates" and the risks of individualized rewards (lines 114) but fails to compare against methods built to solve these exact problems, such as HAPPO [1], MAT [2], A2PO [3], and Partial Reward Decoupling (PRD) [4,5].

  - The problem could also be framed as one of reward redistribution, but relevant methods (e.g., AREL [6], STAS [7], TAR$^2$ [8, 9]) are not discussed or compared.

  - The novelty claim regarding the "transfer of tangible, task-critical resources" (line 123) seems to overlook existing work in environments like Overcooked, where agents must pass items to each other.

- **Overstated Conclusions and Contradictory Results**:

  - The paper's conclusion (line 331) that the problem "can be addressed by the standard policy gradient objective, but not fancier trust region mechanisms" (i.e., PPO) is a significant overstatement based on the provided evidence. The failure of MAPPO/IPPO could be due to many factors.

  - There is a direct contradiction in the results of Section 4.2. Lines 295-297 state that "MARL agents (MAPPO, QMIX, COMA) showing total collapse." However, lines 298-299 state that "only MAPPO and decentralized PG showed any learning in the task." These two statements cannot both be true.

- **Simplicity of the Environment**: The Manitokan task is a very small and simple grid world. The failure of strong, modern baselines like MAPPO on such a trivial task is surprising. This may suggest that the failure is not due to a fundamental "hidden gift" challenge, but rather to the extreme POMDP assumptions (Weakness #2) or insufficient hyperparameter tuning. The paper's claims would be much stronger if demonstrated in a more complex environment.

- **Invalid Theoretical Analysis**: The "Self Correction" term is presented as a formally derived component, but the analysis in Section 5 and Appendix P appears to be an ad-hoc justification for a heuristic. The derivation is based on a series of steps that are not easy to follow (seems flawed) and invalid assumptions.

  - **Appendix P.1 (Deriving the Correction Term)**: This proof is invalid. It makes an incorrect independence assumption to split the expectation in Eq. 13-14 (the score function $\nabla \log \pi$ and the Q-value $Q_c$ are not always independent. Under what conditions are they independent?). Furthermore, the derivation does not actually produce the new Hessian term; the term is simply inserted into the final objective (Eq. 22) without a valid preceding derivation.

  - **Appendix P.2 (Deriving the "Self" Correction)**: This "proof," which is critical for the algorithm's decentralization, is fundamentally flawed. The authors attempt to justify replacing the cross-agent Hessian with a self-Hessian by claiming $\mathbb{E}[Q_c^i] \approx \mathbb{E}[Q_c^j]$. This is a consequence. Proving that both agents' expected collective rewards are related to the same global $r^c$ does not in any way prove that their respective gradient/Hessian terms are equivalent.

  - **Confusing Claims and Typos**: The analysis is further muddied by confusing, unsupported claims (e.g., the "inverse entropy" relationship, line 375) and clear typos (e.g., the missing summation in Equation 8, line 1914), which makes the entire section lack rigor.

[1] Kuba, Jakub Grudzien, et al. "Trust region policy optimisation in multi-agent reinforcement learning." arXiv preprint arXiv:2109.11251 (2021).

[2] Wen, Muning, et al. "Multi-agent reinforcement learning is a sequence modeling problem." Advances in Neural Information Processing Systems 35 (2022): 16509-16521.

[3] Wang, Xihuai, et al. "Order matters: Agent-by-agent policy optimization." arXiv preprint arXiv:2302.06205 (2023).

[4] B. Freed, A. Kapoor, I. Abraham, J. Schneider and H. Choset, "Learning Cooperative Multi-Agent Policies With Partial Reward Decoupling," in IEEE Robotics and Automation Letters, vol. 7, no. 2, pp. 890-897, April 2022

[5] Kapoor, Aditya, et al. "Assigning credit with partial reward decoupling in multi-agent proximal policy optimization." arXiv preprint arXiv:2408.04295 (2024).

[6] Xiao, Baicen, Bhaskar Ramasubramanian, and Radha Poovendran. "Agent-temporal attention for reward redistribution in episodic multi-agent reinforcement learning." arXiv preprint arXiv:2201.04612 (2022).

[7] Chen, Sirui, et al. "STAS: spatial-temporal return decomposition for solving sparse rewards problems in multi-agent reinforcement learning." Proceedings of the AAAI Conference on Artificial Intelligence. Vol. 38. No. 16. 2024.

[8] Kapoor, Aditya, et al. "Agent-Temporal Credit Assignment for Optimal Policy Preservation in Sparse Multi-Agent Reinforcement Learning." arXiv preprint arXiv:2412.14779 (2024).

[9] Kapoor, Aditya, et al. "$ TAR^ 2$: Temporal-Agent Reward Redistribution for Optimal Policy Preservation in Multi-Agent Reinforcement Learning." arXiv preprint arXiv:2502.04864 (2025).

**Questions:**

- **Clarity on Observability (W1)**: Could you please precisely define what is "hidden" in the Manitokan task? When agent $i$ drops the key, is the key's new location on the ground immediately observable in agent $j$'s partial observation (if agent $j$ is looking in that direction)?

- **Clarity on Value Function Inputs (W1)**: For the centralized training baselines (MAPPO, COMA, QMIX), are the centralized critics/value functions conditioned on local action-observation histories or the full global state? If they use the global state, how can the "gift" be considered hidden during training?

- **Success Rate vs. Reward**: In Section 4.1 and Appendix E.4, you note that randomizing the policy can improve the collective success rate but reduce the cumulative reward. What does this imply about the alignment of the cumulative reward objective (which includes individual rewards) and the desired collective task?

- **Basis for Claim (Sec 4.1)**: The claim that agents in MAPPO/IPPO "were still successfully opening their individual doors" (lines 254-2555) is used to explain their higher cumulative reward. Is this a speculation, or is it an observation from evaluating the converged policies (e.g., what percentage of the time did they open their own door)?

- **Contradiction (Sec 4.2)**: Could you please resolve the contradiction in Section 4.2 regarding MAPPO's performance (lines 295-297)? Did it show "total collapse" or "any learning"?

---

> ### Author Response · Authors · 2025-11-25
> **Ethics Concerns**
>
> Dear Reviewer ZmqS,
>
> We are currently preparing our rebuttal and would like to address the ethics flag under “Discrimination / bias / fairness” in a precise way. However, no specific concern related to this principle was articulated in the review. Could Reviewer ZmqS please indicate which aspect of the work they saw as raising this concern, so that we can respond to it appropriately in our rebuttal?

---

> ### Author Response · Authors · 2025-11-28
> **Rebuttal**
>
> ## Weaknesses:
> >Unclear Definition of "Hidden Gift": The central premise of the "hidden gift" is unclearly defined and potentially flawed. The paper states "the act of dropping the key is not actually observable" (line 189). However, it is not easy to know the experimental setup if:
> >* (a) Only the action $a_{drop}$  is hidden from other agents' observations , but the resulting state change (i.e., the key appearing on the ground at ) is observable?
> >* (b) The resulting state change is also hidden (e.g., the key is invisible to agent $j$  even after agent $i$  picks it up)?
>
> >If (a), then this is not a new "hidden gift" problem but a standard MARL problem of non-stationarity and multi-step temporal credit assignment, where the state change is induced by another agent's policy. If (b), this is an extremely strong partial-observability assumption that needs much clearer justification.
>
> It is important to note that this weakness presented from Reviewer ZmqS is constructed as a false dichotomy where the existence of “the hidden gift problem” is assumed false in either of the conditions presented.
>
> In our setting, agents cannot observe each other’s actions or policies directly. When agent 1 drops a key, agent 2 can observe the resulting change in its local state (e.g., the key was absent and now it is present), but this is only the consequence of some unobserved action chosen by agent 1’s policy. Agent 2 does not see that agent 1 intentionally chose to “gift” the key, nor whether agent 1 will behave this way again in the future, so the gift itself is effectively hidden. With memory, agent 2 can in principle infer that the dropped key is intended for it to pick up and use to open its own door for the collective reward, since agent 1 should not have dropped the key before securing its own individual reward unless it was acting prosocially. The core challenge is therefore to infer such a “hidden” gift from agent 1 and to learn to reciprocate when the roles are reversed.
>
> There is partial observability in the challenge of hidden gifts, since it is a Dec-POMDP, but the extent of  how partial is standard and aligned with past MARL work that we expand on below.
>
> > Extreme and Unjustified Partial Observability: The initial experiments (Sec 4.1) appear to be conducted under extreme partial observability, where "agents did not have an explicit signal for their door being opened or that they are holding the key" (lines 268-270). This lack of basic proprioceptive state (knowing if one is holding the key or has completed one's own task) makes the problem artificially difficult and is a highly unusual assumption for MARL environments. This should be made far clearer and be justified.
>
> To elaborate on what we meant by a “lack of door or key signal”: in the initial experiments there is no dedicated observation feature at every time step that explicitly indicates whether an agent is holding the key or whether its door is open. However, the observation space does contain enough information, together with policy memory, for a learned policy to select the correct actions at each state and solve the task. We discussed this design in detail with Reviewers 5tWK and TeS5 when formalizing the Dec-POMDP.
> The wording in the review (using terms such as “extreme,” “unjustified,” “basic,” “artificially,” and “unusual”) can be read as implying that we intentionally contrived an unrealistic problem rather than discovering one. It is not the case we made an artificially difficult task. Our initial environment follows a Dec-POMDP setup in which agents receive only local observations.
>
> We also do not view partial observability as lying on a meaningful gradient of “extremity”: either an environment is fully observable or it is partially observable. Prior MARL work, including STAS (the task cited by the reviewer), has already shown that pure partial observations are sufficient to learn the task (see https://web.media.mit.edu/~cynthiab/Readings/tan-MAS-reinfLearn.pdf). The Manitokan task follows this tradition: we deliberately avoid adding global state variables that trivialize and change the inference problem, while also providing experiments with decentralized state features (including explicit door/key information) that could have reasonably improved performance. This design choice, and how it differs from MARL environments that exploit more global state information, is discussed in the related work, and we will clarify it further in the revision.

---

> > ### Author Response · Authors · 2025-11-28
> > **Continued Rebuttal**
> >
> > > Missing Key Baselines and Related Work: The paper's primary claim is that existing MARL algorithms fail at this credit assignment problem. However, it omits a large and highly relevant body of work specifically designed to address non-stationarity and credit assignment in CTDE settings.
> >
> > > * The paper notes issues like "conflicting gradient updates" and the risks of individualized rewards (lines 114) but fails to compare against methods built to solve these exact problems, such as HAPPO [1], MAT [2], A2PO [3], and Partial Reward Decoupling (PRD) [4,5].
> >
> > It is correct that these new baselines address non-stationarity and credit assignment for common games like all of the other algorithms we evaluated but they do not appear to address the “hidden gift” problem itself. Some even appear incompatible with the Manitokan task due to limitations stated in their papers.
> >
> > - Partial Reward Decoupling (PRD): In the limitation section of the original PRD paper, it is stated that PRD cannot work in games with collective rewards. So we are perplexed as to why the reviewers proposed this method as a solution since the Manitokan task has a collective reward. In the new Partial Reward Decoupling paper, PRD-MAPPO is able to decouple shared rewards into individual rewards, but the decoupled rewards must sum to the global reward. It is not immediate how this handles the non-stationary collective reward with the stationary individual reward of the Manitokan task. However, we will consider this as a new baseline.
> >
> > - MAT: In our limitation section, we discussed the possible limitations of GRU memory and how an autoregressive network on a longer context may be more sample efficient in learning hidden gifts. Both SAF and QTRAN have transformer networks for meta learning policies and computed a global value function respectively but did not improve performance. We will consider running MAT as a new baseline.
> >
> > - HAPPO and A2PO: We tested MAPPO, IPPO and SAF which are SOTA PPO variants in MARL tasks with SAF leveraging a meta learning architecture to improve performance. HAPPO and the subsequent A2PO are new training methods for Multi-agent PPO that segment which agent gets updated to avoid inter agent gradient noise or agent local optimal updates inhibiting agent global optimal updates. We will consider running A2PO as a new baseline.
> >
> > > * The problem could also be framed as one of reward redistribution, but relevant methods (e.g., AREL [6], STAS [7], TAR [8, 9]) are not discussed or compared.
> >
> > Although the Reviewer is correct in that any reinforcement learning problem can be framed as a reward redistribution problem, we did investigate a similar meta-learning technique which does act as a solution to the problems TAR, STAS and AREL solve. Stateful Active Facilitator (SAF), during training, incorporates a latent knowledge source where agents share information that is then compressed then sent back to train the critic and a policy pool so agents can dynamically select a policy at their current state. This is similar to transformer based reward redistribution models up until the computation of a reward where SAF computes a policy for agents instead. We will consider TAR as a baseline.
> >
> > > * The novelty claim regarding the "transfer of tangible, task-critical resources" (line 123) seems to overlook existing work in environments like Overcooked, where agents must pass items to each other.
> >
> > We did mention Overcooked in the related works section but we did not state the particular nuance between the Manitokan task and Overcooked scenarios. The only scenario in overcooked is the Forced Coordination scenario where agents are in separate areas and one has to pass ingredients across a table for the other to then place in the oven. This is an example of transfer of resources but there is a lack of scarcity and reuse of the resource which we only mention in the Manitokan tasks section. It could be possible to reskin the Manitokan task as an overcooked environment where two chefs need to share a lighter for their ovens but this scenario does not exist in any overcooked suite to our knowledge. We will add this nuance in the related work section.

---

> > > ### Author Response · Authors · 2025-11-28
> > >
> > > > Overstated Conclusions and Contradictory Results:
> > >
> > > > * The paper's conclusion (line 331) that the problem "can be addressed by the standard policy gradient objective, but not fancier trust region mechanisms" (i.e., PPO) is a significant overstatement based on the provided evidence. The failure of MAPPO/IPPO could be due to many factors.
> > >
> > > We will change this to say, “but not fancier trust region mechanisms we tested namely MAPPO, IPPO and SAF”.
> > >
> > > > * There is a direct contradiction in the results of Section 4.2. Lines 295-297 state that "MARL agents (MAPPO, QMIX, COMA) showing total collapse." However, lines 298-299 state that "only MAPPO and decentralized PG showed any learning in the task." These two statements cannot both be true.
> > >
> > > Thank you for pointing this out. MAPPO’s collective success rate does collapse but its learning does not. This is verified in the reward plots.. We will add context to these lines but they are not contradictory.
> > >
> > > > Simplicity of the Environment: The Manitokan task is a very small and simple grid world. The failure of strong, modern baselines like MAPPO on such a trivial task is surprising. This may suggest that the failure is not due to a fundamental "hidden gift" challenge, but rather to the extreme POMDP assumptions (Weakness #2) or insufficient hyperparameter tuning. The paper's claims would be much stronger if demonstrated in a more complex environment.
> > >
> > > We agree that it is surprising that strong, modern baselines such as MAPPO fail on a small gridworld. However, this is precisely one of the key messages of our work: methods that perform well on standard benchmarks do not automatically generalize to the specific challenge posed by hidden gifts, even after we carefully tuned their hyperparameters.
> > >
> > > Although the Manitokan environment looks simple, it is deceptively hard for RL agents because it requires assigning credit between a non-stationary collective reward and stationary individual rewards under partial observability. As the Reviewer ZmqS has noted previously, the task is in fact difficult for current methods despite its small state space.
> > >
> > > We deliberately use a small gridworld to make the problem analytically and computationally tractable: it allows us to isolate the hidden-gift phenomenon, run extensive training and hyperparameter sweeps, and avoid confounding factors introduced by large-scale, visually complex environments. Demonstrating similar effects in richer environments would indeed be interesting, but that extension is beyond the scope of this paper and is a natural direction for future work.
> > >
> > > > Invalid Theoretical Analysis: The "Self Correction" term is presented as a formally derived component, but the analysis in Section 5 and Appendix P appears to be an ad-hoc justification for a heuristic. The derivation is based on a series of steps that are not easy to follow (seems flawed) and invalid assumptions.
> > >
> > > > - Appendix P.1 (Deriving the Correction Term): This proof is invalid. It makes an incorrect independence assumption to split the expectation in Eq. 13-14 (the score function  and the Q-value  are not always independent. Under what conditions are they independent?). Furthermore, the derivation does not actually produce the new Hessian term; the term is simply inserted into the final objective (Eq. 22) without a valid preceding derivation.
> > >
> > > The conditions of independence are explicitly stated in the sketch of the proof as well as the derivation. To reiterate, the policy of agent 1 can not sample an action to maximize the collective reward because the collective reward only requires agent 2 therefore the score function of agent 1 is independent of agent 1’s collective Q value estimate. The first differentiation comes from using agent 2’s objective to derive the correction term as a surrogate for the collective reward. The next differentiation comes from differentiating with respect to the agent 1 after the correction term replaces the collective reward in the lookahead step. Since one needs to differentiates twice to get the correction term, it is a Hessian.

---

> > > > ### Author Response · Authors · 2025-11-28
> > > > **Final Rebuttal**
> > > >
> > > > > - Appendix P.2 (Deriving the "Self" Correction): This "proof," which is critical for the algorithm's decentralization, is fundamentally flawed. The authors attempt to justify replacing the cross-agent Hessian with a self-Hessian by claiming . This is a consequence. Proving that both agents' expected collective rewards are related to the same global  does not in any way prove that their respective gradient/Hessian terms are equivalent.
> > > >
> > > > The reviewer seems to have misread the relation. The full relation is $\mathbb{E}_{\tau\sim\pi^i}[\nabla_{\Theta^i}J_c(\Theta^i)\Psi(\pi_c^i,a^i,o^i)]=\mathbb{E}_{\tau\sim\pi^j}[Q_c(o^j,a^j)] \approx\mathbb{E}_{\tau\sim\pi^j,\tau\sim\pi^i}[ r^c]\approx \mathbb{E}_{\tau\sim\pi^i}[Q_c(o^i,a^i)] = \mathbb{E}_{\tau\sim\pi^i}[\nabla_{\Theta^i}J_c(\Theta^i)\Psi(\pi_c^i,a^i,o^i)]$ between equations 25-28. This does prove that the correction terms are equivalent in expectation since they both approximate the collective reward under the context outline by the reward function. As a reminder, the reward function’s conditions states that the gifting agent must have opened their door first which is the context where the correction terms exist.
> > > >
> > > > > - Confusing Claims and Typos: The analysis is further muddied by confusing, unsupported claims (e.g., the "inverse entropy" relationship, line 375) and clear typos (e.g., the missing summation in Equation 8, line 1914), which makes the entire section lack rigor.
> > > >
> > > > Mathematically, the definition of entropy is the expectation of surprisal $\log p$. This is not a claim, but a fact. Since the policy is a probabilistic network then $\log( \pi)$ is the surprisal of that network and the expectation of that surprisal is the  entropy of that network. Inverting the expectation of this is the inverse entropy of that network. We will omit this description from the work as a certain level needs to be assumed before engaging with our paper.
> > > >
> > > > We disagree that our stylistic choice to omit the summation in Eq (8). renders the section to lack rigour. We added the summations back in the revised paper.
> > > >
> > > > ## Questions:
> > > > > Clarity on Observability (W1): Could you please precisely define what is "hidden" in the Manitokan task? When agent  drops the key, is the key's new location on the ground immediately observable in agent 's partial observation (if agent  is looking in that direction)?
> > > >
> > > > We clarified this in the first weakness.
> > > >
> > > > > Clarity on Value Function Inputs (W1): For the centralized training baselines (MAPPO, COMA, QMIX), are the centralized critics/value functions conditioned on local action-observation histories or the full global state? If they use the global state, how can the "gift" be considered hidden during training?
> > > >
> > > > The critics are trained on the agent’s observation rather than the full global state.
> > > >
> > > > > Success Rate vs. Reward: In Section 4.1 and Appendix E.4, you note that randomizing the policy can improve the collective success rate but reduce the cumulative reward. What does this imply about the alignment of the cumulative reward objective (which includes individual rewards) and the desired collective task?
> > > >
> > > > This implies the cumulative reward has difficulty properly assigning credit to maximize the aforementioned objective and needs a correction term to better predict the collective reward. This further speaks to the challenge of hidden gifts and interesting results of this work.
> > > >
> > > > > Basis for Claim (Sec 4.1): The claim that agents in MAPPO/IPPO "were still successfully opening their individual doors" (lines 254-2555) is used to explain their higher cumulative reward. Is this a speculation, or is it an observation from evaluating the converged policies (e.g., what percentage of the time did they open their own door)?
> > > >
> > > > This is an observation from evaluating the converged policies. This can be directly inferred by the cumulative reward plots since both the individual and collective require at least one agent’s door to be opened.
> > > >
> > > > > Contradiction (Sec 4.2): Could you please resolve the contradiction in Section 4.2 regarding MAPPO's performance (lines 295-297)? Did it show "total collapse" or "any learning"?
> > > >
> > > > This is not a contradiction as we mentioned above. We will elaborate that MAPPO’s collective success rate collapses but learning does not to avoid confusion.

---

### Author Response · Authors · 2025-11-28

Dear Reviewers,

We would like to sincerely thank you for your thoughtful and detailed feedback, which prompted deep self-reflection on our work and much deliberation amongst authors. In particular, your comments led to several concrete improvements in the paper.

The discussion around the correct game formalization pushed us to clarify the Manitokan task as a fully cooperative Dec-POMDP with a monotonic reward structure, and to explain more precisely how the individual and collective reward components interact for a single global objective. While the critiques over the theoretical section motivated us to sharpen the presentation of the key assumptions and steps in the derivation, to state expectations and independence conditions explicitly, and to better highlight what is unique about the correction terms relative to past work in this hidden-gift setting.

Lastly, the questions about what information agents actually use led us to add new interesting experiments showing that:
- A single last action in the current observation is often sufficient for learning (appendix E.10)
- The temporal structure between actions is crucial (appendix E.10)
- The self-correction term not only stabilizes learning but also yields globally higher collective success rates than the other policy-gradient models (appendix E.12)

These clarifications and additional results sharpen our contributions and better highlight the distinctive novelty of the challenge of hidden gifts.

---

### Note · Authors · 2026-01-21

**Comment:**

There is a notational error in equation 3 and the appendix derivations that render the submission unfit for publication. This error does not exist in the submitted python nor in the empirical results.

The correction terms should not have the gradient in front of the scored policy but in front of the q-value estimates of the collective reward $\\mathbb{E}\_{\\tau\\sim\\pi^j}[\\log\\pi^i_c(a^i|o^i)\\nabla\_{\Theta^j}Q^i_c(a^i,o^i)]$.

So, using the collective reward in lieu of the q-value estimate of it, in an expansion of equation 2:

$$
J(\Theta^i)
$$

$$
=\\mathbb{E}\_{\\tau\\sim\\pi^i}[\\sum\_{t=0}^T\\gamma^t\\hat{\\mathcal{R}}^i(o\_t^i,a\_t^i)]
$$

$$
=\\mathbb{E}\_{\tau\sim\pi^i}[\sum\_{t=0}^T \gamma^tr\_t^i+ \gamma^tr\_t^c]
$$

$$
=\\mathbb{E}\_{\tau\sim\pi^i}[\\sum\_{t}^{T-1}\\gamma^t r\_t^i] + \\mathbb{E}\_{\tau\sim \pi^i}[\\sum\_{t=T}^T \gamma^tr\_t^c]
$$

$$
=\\mathbb{E}\_{\\tau\\sim\\pi^i}[\\sum\_{t=0}^{T-1} \\gamma^tr\_t^i] + \\mathbb{E}\_{\\tau\\sim \\pi^i}[ \\gamma^Tr\_T^c]
$$

Per the reward function, both the collective and individual rewards can appear at most once per episode, with the collective reward occurring only at the terminal time step. Therefore, the summation is dropped as it only sums at the terminal time step $T$.
<br>
Then the objective for agent $i$'s collective reward is the expected collective reward at termination:

$$J\_c(\\Theta^i)
= J(\\Theta^i)- \\mathbb{E}\_{\\tau\\sim \\pi^i}\\left[\\sum\_{t=0}^{T-1} \\gamma^t r\_t^i\\right]
= \\mathbb{E}\_{\\tau\\sim \\pi^i}\\left[\\gamma^T r\_T^c\\right].
$$

Moreover, since the collective reward $r_T^c$ is assumed to have been returned, then the other agent $j$ must have completed the task. The collective reward, from the last timestep, is conditioned on agent $j$'s policy and therefore  agent $i$'s objective is  $J\_c(\\Theta^i)=\\mathbb{E}\_{\\tau\\sim\\pi^j}\\left[\\log \\pi^j\_c(a^j|o^j)\\gamma^T r\_T^c\\right]$. That is, the collective reward is weighted by scored policy. This is commonly what is constructed during training before backpropping it's gradient.

Now if we differentiate the above term with respect to agent $i$'s parameters,

$\nabla\_{\Theta^i}J\_c(\\Theta^i)=\nabla\_{\Theta^i}\\mathbb{E}\_{\\tau\\sim\\pi^j}\\left[\\log \\pi^j\_c(a^j|o^j)\\gamma^T r\_T^c\\right]$

$=\\nabla\_{\\Theta^i}\\sum\_{a^j\\in\\mathcal{A}}\\pi^j(a^j|o^j)\\log \\pi^j\_c(a^j|o^j)\\gamma^Tr^c\_T]$

$=\\sum\_{a^j\\in\\mathcal{A}}\\pi^j(a^j|o^j)\\log \\pi^j\_c(a^j|o^j)\\nabla\_{\\Theta^i}\\gamma^Tr^c\_T$

Now to show that the actions between agent's are independent of each other, we can use the identity of  the policy gradient theorem

$= \\sum\_{a^j\\in\\mathcal{A}}\\pi^j(a^j|o^j)\\log \\pi^j\_c(a^j|o^j)\\nabla\_{\\Theta^i}\\log\\pi^i\_c(a^i|o^i)r^c\_T$

Then we have a _conditional independence_ between the agent $i$'s term and agent $j$'s term. This independence is guaranteed because the collective reward was returned which means the key was "gifted" which is the conditioning variable.

$= (\\sum\_{a^j\\in\\mathcal{A}}\\pi^j(a^j|o^j)\\log \\pi^j\_c(a^j|o^j))(\\sum\_{a^j\\in\\mathcal{A}}\\pi^j(a^j|o^j)\\nabla\_{\\Theta^i}\\log\\pi^i\_c(a^i|o^i)r^c\_T)$

$= \\mathbb{E}\_{\\tau\\sim\\pi^j\_c}[\\log \\pi^j\_c(a^j|o^j)]\\mathbb{E}\_{\\tau\\sim\\pi^j\_c}[\\nabla\_{\\Theta^i}\\log\\pi^i\_c(a^i|o^i)r^c\_T] $

$= \\mathbb{E}\_{\\tau\\sim\\pi^j\_c}[\\log \\pi^j\_c(a^j|o^j)]\\mathbb{E}\_{\\tau\\sim\\pi^j\_c}[\\nabla\_{\\Theta^i}\\gamma^Tr^c\_T]$

Then the final correction term is $$\\frac{\\nabla\_{\\Theta^i}J\_c}{\\mathbb{E}\_{\\tau\\sim\\pi^j\_c}[\\log \\pi^j\_c(a^j|o^j)]}=]\\mathbb{E}\_{\\tau\\sim\\pi^j\_c}[\\nabla\_{\\Theta^i}\\gamma^Tr^c\_T]$$ which is used as a surrogate in the look ahead step of the differentiated collective reward in the policy gradient derivation.

Notably $\\nabla\_{\\Theta^i}J\_c\\rightarrow0$ and $\\mathbb{E}\_{\\tau\\sim\\pi^j\_c}[\\nabla\_{\\Theta^i}\\gamma^Tr^c\_T]\\rightarrow0$ since, as agents learn, there should be no effect of an agents' policy on the collective reward.

$$\\nabla\_{\\Theta^i}J\_c={\\mathbb{E}\_{\\tau\\sim\\pi^{*j}\_c}[\\log \\pi^{*j}\_c(a^j|o^j)]}\\mathbb{E}\_{\\tau\\sim\\pi^{*j}\_c}[\\nabla\_{\\Theta^i}\\gamma^Tr^c\_T]=0={\\mathbb{E}\_{\\tau\\sim\\pi^{*i}\_c}[\\log \\pi^{*i}\_c(a^i|o^i)]}\\mathbb{E}\_{\\tau\\sim\\pi^{*i}\_c}[\\nabla\_{\\Theta^j}\\gamma^Tr^c\_T]=\\nabla_{\\Theta^j}J\_c$$

where $\\pi^*$ is an optimal policy.

The variables can be manipulated since they degenerate to zero with an optimal policy

$$\\nabla\_{\\Theta^i}J\_c={\\mathbb{E}\_{\\tau\\sim\\pi^{*i}\_c}[\\log \\pi^{*i}\_c(a^i|o^i)]}\\mathbb{E}\_{\\tau\\sim\\pi^{*i}\_c}[\\nabla\_{\\Theta^j}\\gamma^Tr^c\_T]=0={\\mathbb{E}\_{\\tau\\sim\\pi^{*j}\_c}[\\log \\pi^{*j}\_c(a^j|o^j)]}\\mathbb{E}\_{\\tau\\sim\\pi^{*j}\_c}[\\nabla\_{\\Theta^i}\\gamma^Tr^c_T]=\\nabla\_{\\Theta^j}J\_c$$

The self-correction comes from:

$$\\frac{\\nabla\_{\\Theta^i}J\_c}{\\mathbb{E}\_{\\tau\\sim\\pi^{*i}\_c}[\\log \\pi^{*i}\_c(a^i|o^i)]}=\\mathbb{E}\_{\\tau\\sim\\pi^{*i}\_c}[\\nabla\_{\\Theta^j}\\gamma^Tr^c\_T]=0=\\mathbb{E}\_{\\tau\\sim\\pi^{*j}\_c}[\\nabla\_{\\Theta^i}\\gamma^Tr^c\_T]=\\frac{\\nabla\_{\\Theta^j}J\_c}{\\mathbb{E}\_{\\tau\\sim\\pi^{*j}\_c}[\\log \\pi^{*j}\_c(a^j|o^j)]}$$

The use of an agent's own policy to predict the change in collective reward sampled is what we called "self"-correction, a *self-learning awareness* term rather than a collective learning awareness term.

**Withdrawal Confirmation:**

I have read and agree with the venue's withdrawal policy on behalf of myself and my co-authors.